# Unraveling the thermodynamic criteria for size-dependent spontaneous phase separation in soft porous crystals

Sven M.J. Rogge [1], Michel Waroquier[1] & Veronique Van Speybroeck [1*]

Soft porous crystals (SPCs) harbor a great potential as functional nanoporous materials owing to their stimuli-induced and tuneable morphing between different crystalline phases. These large-amplitude phase transitions are often assumed to occur cooperatively throughout the whole material, which thereby retains its perfect crystalline order. Here, we disprove this paradigm through mesoscale first-principles based molecular dynamics simulations, demonstrating that morphological transitions do induce spatial disorder under the form of interfacial defects and give rise to yet unidentified phase coexistence within a given sample. We hypothesize that this phase coexistence can be stabilized by carefully tuning the experimental control variables through, e.g., temperature or pressure quenching. The observed spatial disorder helps to rationalize yet elusive phenomena in SPCs, such as the impact of crystal downsizing on their flexible nature, thereby identifying the crystal size as a crucial design parameter for stimuli-responsive devices based on SPC nanoparticles and thin films.

---

[1] Center for Molecular Modeling (CMM), Ghent University, Technologiepark 46, 9052 Zwijnaarde, Belgium. *email: Veronique.VanSpeybroeck@UGent.be

Within generic classifications of materials, from soft polymers to stiff metals and ceramics, metal-organic frameworks (MOFs) or porous coordination polymers hold a very peculiar role, since they are built from both inorganic and organic fragments. The labile coordination bond between these fragments, which dominates a MOF's formation and architecture, hampers a straightforward extrapolation of standard material concepts to MOFs. While MOFs were initially thought to possess only very little spatial disorder[1], there is now clear evidence for the presence of intrinsic disorder in these materials[2–8]. The relatively weak interactions governing MOFs[9,10] tolerate spatial heterogeneities that vary from point-like defects[11–13] to completely amorphous and glass-like 3D phases lacking any long-range structural order[14–16], impacting the MOF's performance[5,17].

Besides these already known types of spatial heterogeneities, morphological transitions in MOFs may induce an additional and yet-unidentified type of spatial disorder under the form of interfacial defects. Such phenomenon could therefore exist in soft porous crystals (SPCs) or flexible MOFs[9,18], which show large-amplitude transitions between different (meta)stable phases while retaining their crystallinity[19–21]. When assuming perfect crystals, these transitions would occur cooperatively[20,22], so that an SPC can only exhibit a single phase at any given time[23]. However, recent experiments evidenced that downsizing SPCs from the micro- to the mesoscale—with primary crystallite sizes between 10 nm and 1 μm, see Fig. 1c—substantially suppresses their structural mobility. These size-dependent phenomena, including the absence of negative gas adsorption in DUT-49(Cu) below a critical crystal size[24] and suppressed phase transitions in meso-sized pillared-layered SPCs[25–28], contradict the assumption of collective flexibility.

Herein, we show that the key to unravel the transition mechanism is to explore larger MOF cells that are several tens of nanometers in size. Using these mesocells, which are more than one order of magnitude larger than the traditionally considered nanocells (see Fig. 1c), we reveal that multiple phases can coexist in various pillared-layered SPCs, giving rise to interfacial defects (see Fig. 1a, b). The observed phase coexistence depends on the crystallite size and is stabilized by judiciously tuning pressure, temperature, or guest adsorption. Based on these insights, we here hypothesize different pathways to experimentally observe phase coexistence in SPCs, paving the way to leverage spatially disordered SPCs for targeted applications.

## Results

**Size-dependent phase transitions in SPCs**. Few pillared-layered SPCs have received as much attention as MIL-53 (MIL = Matérial de l'Institut Lavoisier)[29]. Its winerack structure endows MIL-53 with the ability to dynamically morph between two metastable phases, the closed-pore (cp) and large-pore (lp) phase, when subjected to changes in mechanical pressure or temperature. A third phase with intermediate volume, the narrow-pore (np) phase, can be accessed through guest adsorption. Although the MIL-53 phase transitions have been investigated by numerous theoretical studies, they were inherently limited to small MOF cells that we will here refer to as nanocells. Very recently, however, Kundu et al. observed that the dynamic behavior of MIL-53(Al)-NH$_2$ is size-dependent, which was used to optimize the material for natural gas delivery[28]. To understand the nature of this size-dependent flexibility, we systematically investigate the response of the parent material, MIL-53(Al), to a pressure of 40 MPa at 300 K, as the material breathes experimentally under

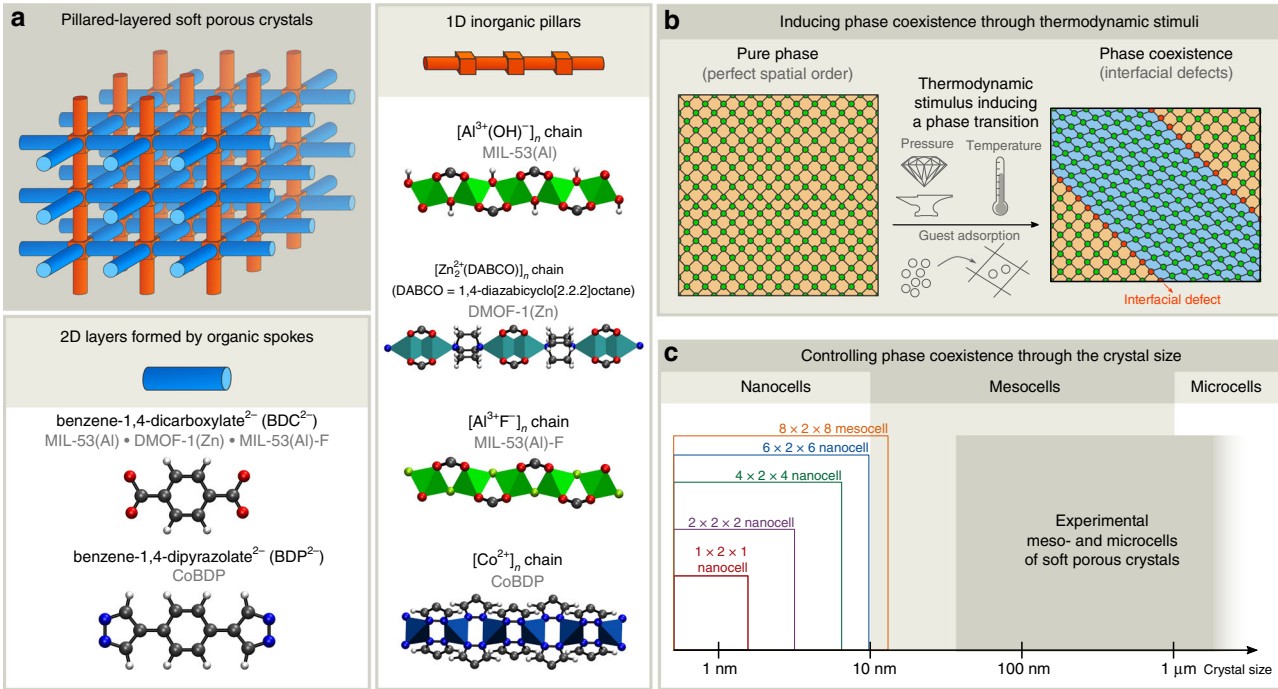

**Fig. 1** Inducing size-dependent phase coexistence in pillared-layered SPCs using various thermodynamic stimuli. **a** Atomistic representation of the building blocks of the SPCs discussed in this work: MIL-53(Al), DMOF-1(Zn), MIL-53(Al)-F, and CoBDP. Color code: zinc (cyan), cobalt (cobalt blue), aluminum (green), fluor (lime), oxygen (red), nitrogen (blue), carbon (gray), hydrogen (white). **b** The three investigated thermodynamic stimuli to induce phase coexistence: pressure, temperature, and guest adsorption. The SPC's metal nodes act as hinges which are either in equilibrium (green) or locally deformed (red). **c** Classification of the theoretical and experimental crystal sizes into nanocells, mesocells, and microcells. The color code for the theoretical nano- and mesocells is used consistently throughout this work

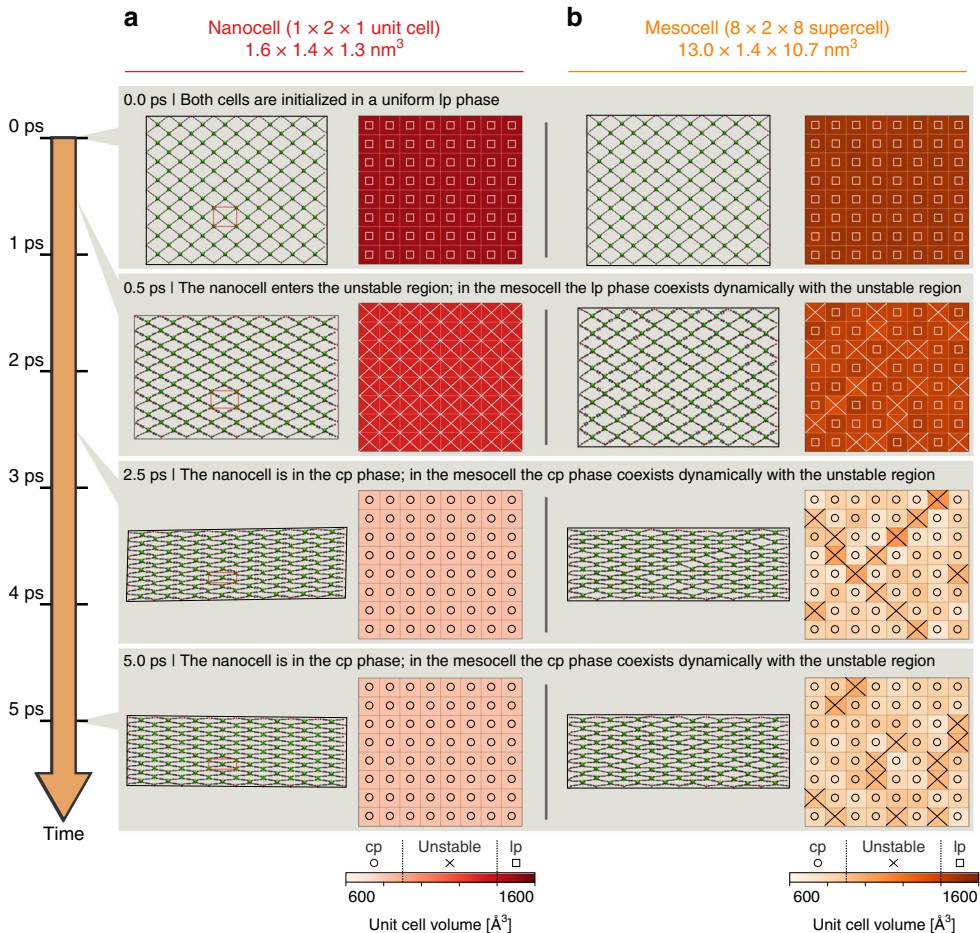

**Fig. 2** Size-dependent mechanism underpinning the phase transition in MIL-53(Al). Visualization of the lp-to-cp transition in MIL-53(Al) during a constant-temperature constant-pressure simulation at 300 K and 40 MPa, using **a** a nanocell (1 × 2 × 1 unit cell), which is indicated with a red rectangle and periodically repeated to match the size of the mesocell, and **b** a mesocell (8 × 2 × 8 unit cell). The atomic visualizations are supplemented by schematic representations that are color coded based on the volume of the different subcells (structural classification). Furthermore, the mechanically stable closed-pore (cp) and large-pore (lp) phases and the intermediate unstable region are indicated by circles, squares, and crosses, respectively (mechanical classification). Due to periodic boundary conditions, spatial disorder is only tolerated within the simulation cell

these thermodynamic conditions[30,31]. To this end, the different cell sizes indicated in Fig. 1c, varying from the traditional 1 × 2 × 1 nanocell (1.6 × 1.4 × 1.3 nm³) to the extended 8 × 2 × 8 mesocell (13.0 × 1.4 × 10.7 nm³), were considered (the direction parallel to the inorganic chain does not influence the flexibility, see Supplementary Note 4), assuming periodic boundary conditions (see Methods). Although the mesocells are still about an order of magnitude smaller than experimental mesocells (see Fig. 1c), they already reveal new physicochemical phenomena that are suppressed in nanocells.

In Fig. 2, the mechanisms for the lp-to-cp transition of the MIL-53(Al) nanocell and mesocell are contrasted (see Supplementary Movies for other cell sizes). These observations result from following MIL-53(Al) during an isothermal-isobaric simulation at 300 K and 40 MPa. The subcells (see Supplementary Note 3) are marked with squares and circles to indicate the (meta) stable lp and cp phase, respectively, whereas crosses indicate the mechanically unstable volume region that separates both phases. For the nanocell, the lp-to-cp transition occurs cooperatively, as the periodic boundary conditions prevent any spatial disorder between adjacent nanocells. For the mesocell, the transition time differs due to the larger structure, but, more importantly, the transition itself also propagates differently through the material. When starting from the same initial structure, but allowing for

disorder within the mesocell, the different subcells are no longer strictly correlated. Upon exposure to a pressure of 40 MPa at 300 K, some parts of the mesocell undergo an lp-to-cp transition first, thereby crossing the unstable volume range, while other subcells temporarily remain in the lp phase. Correspondingly, the lp-to-cp transition spontaneously introduces transient and dynamic spatial disorder in the mesocell of Fig. 2. Interestingly, although the subcells are not strictly correlated and horizontally and vertically adjacent subcells often exhibit largely different volumes, diagonally adjacent subcells prefer to exhibit similar volumes (see also Supplementary Note 4). This originates from the material's winerack topology, as diagonally adjacent subcells are connected through BDC ligands, leading to the peculiar layer-by-layer breathing in Fig. 2 that was postulated earlier by Triguero et al.[32] and which was also observed in ref. [8] for DMOF-1(Zn).

**Nanoscopic origin of phase coexistence.** To unravel the origin of the observed dynamic disorder, a thermodynamic interpretation is mandatory. To this end, we apply our earlier proposed thermodynamic formalism to construct pressure and free energy equations of state as function of the volume[21,33]. They yield unique insight in the flexible behavior of SPCs, complementing experiments to predict the thermodynamic conditions of temperature,

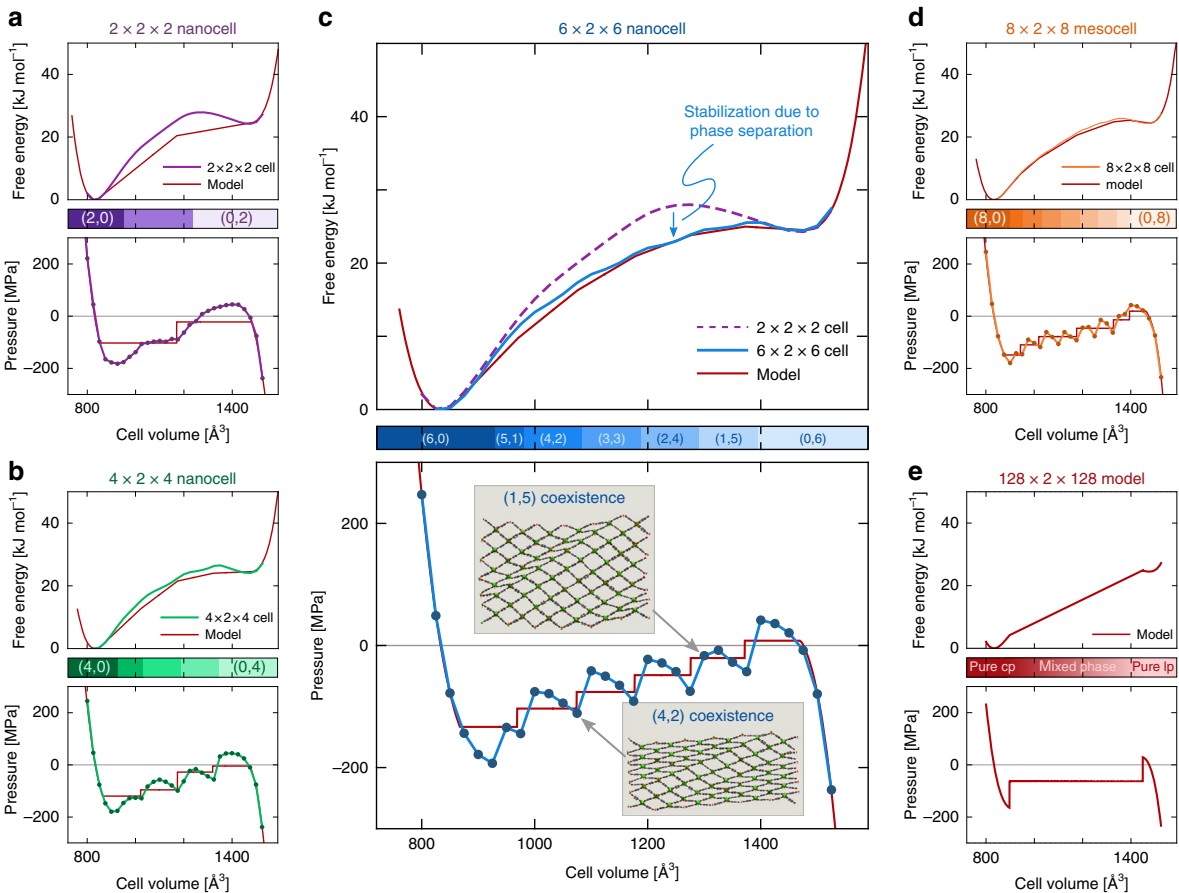

**Fig. 3** Stabilization of the metastable cp/lp coexistence regions with increasing cell size. Pressure (bottom) and free energy (top) equations of state, with indication of the metastable $(n_{cp}, n_{lp})$ phase coexistence regions, in which $n_{cp}$ cp and $n_{lp}$ lp layers coexist, as a function of the volume (middle) for four different cell sizes: **a** a 2 × 2 × 2 supercell, **b** a 4 × 2 × 4 supercell, **c** a 6 × 2 × 6 supercell, and **d** an 8 × 2 × 8 supercell, all at 300 K. The red lines indicate the fitted thermodynamic model, which is extrapolated in **e** for a 128 × 2 × 128 cell (208.0 × 1.4 × 171.2 nm³), similar in size to experimental MIL-53(Al)-NH₂ crystals (see Supplementary Note 2)

pressure, and chemical potential under which a material becomes flexible (see Methods section)[21]. For a given material, its different (meta)stable states correspond to the minima in the obtained free energy profile and a negative slope of the pressure profile, whereas its transition states are found as local maxima and a positive pressure slope. This methodology was extensively tested on nanocells of various materials, including MIL-53(Al)[21,33,34]. Here, the formalism is adopted for both nanocells and mesocells, as shown in Fig. 3. The profiles are clearly sensitive to the considered cell size. When enlarging the simulation cell in which spatial disorder can be captured, the pressure profile becomes increasingly staggered. The lp-to-cp and cp-to-lp transition pressures, which correspond to the local maximum near the lp phase and the local minimum near the cp phase, respectively, remain largely size-independent (see Supplementary Note 5). However, the volume regions between the pure cp and lp phases are fundamentally altered. While the spatially ordered nanocell of Fig. 3a exhibits a mechanically unstable region (positive pressure slope) at intermediate volumes, these intermediate regions become metastable (negative pressure slope) at pressures between about −180 MPa and 0 MPa for the spatially disordered mesocell of Fig. 3d. These negative pressures correspond with pulling the material isotropically, which is not yet experimentally feasible.

To further investigate the nanoscopic nature of these intermediate regions, a 6 × 2 × 6 MIL-53(Al) nanocell is highlighted in Fig. 3c. In this case, five intermediate regions with a negative slope are encountered at volumes between the pure cp (around

820 Å³) and lp (around 1475 Å³) regions. Visualizing the MIL-53(Al) structures along this profile reveals that the intermediate domains correspond to different phase coexistence regions. At the highest volumes reported in Fig. 3c, MIL-53(Al) adopts a pure lp phase. Upon decreasing the volume, one of the layers in the material undergoes an lp-to-cp transition while the other layers remain in the lp phase, thereby introducing two interfacial defects between the cp and lp phase. These interfacial defects, which are suppressed in smaller nanocells, form a yet unidentified type of spatial disorder in SPCs that is accompanied by local deformations at the inorganic node (see Fig. 1b). The observed layer-like behavior stems from the strong correlation between diagonally adjacent subcells already observed in Fig. 2b. As 1 cp and 5 lp layers coexist in this region (see Fig. 3c), it is denoted the $(n_{cp} = 1, n_{lp} = 5)$ coexistence region. This coexistence is dynamic, as the cp layer can propagate throughout the framework. It is postulated that the overall kinetics of the phase transition is correlated with the time scale needed for this wave to propagate through the lattice. When further decreasing the volume, additional layers start to systematically undergo an lp-to-cp transition, until the pure $(n_{cp} = 6, n_{lp} = 0)$ phase is reached, as indicated with the colorbar. The corresponding free energy profile of Fig. 3c indicates that, at a given intermediate volume, phase coexistence stabilizes the material with respect to the models precluding spatial disorder, such as the 2 × 2 × 2 nanocell.

As the observed phase coexistence in pillared-layered SPCs systematically occurs in a layer-by-layer fashion, larger cells allow

for a larger variety of possible phase coexistence regions. To explore how the flexible nature of this SPC changes when further enlarging the crystal, a generally applicable thermodynamic model is derived (see Methods section). This model predicts the pressure and free energy of a given $(n_{cp}, n_{lp})$ phase coexistence region as function of the volume, while only requiring information on the pure lp and cp phases and the cp/lp interfacial free energy barrier. As indicated in Fig. 3, the model (red line) corresponds well with the simulated results starting from a $4 \times 2 \times 4$ cell onwards when assuming an activation barrier of about 11 kJ mol$^{-1}$ $n^{-1/2}$ for the formation of an cp/lp interface, where $n$ is the number of layers in the mesocell (see Supplementary Note 1). The model therefore confirms the experimental observation that the barrier to introduce an interface increases the smaller the crystal[25], a result also obtained independently by Keupp et al.[8]. Given its transferability, our model can be easily adopted to predict the flexibility of even larger cells, such as the $128 \times 2 \times 128$ mesocell with a cell size of $208 \times 1.4 \times 171$ nm$^3$ (see Fig. 3e), similar to experimental MIL-53(Al)-NH$_2$ mesocrystals (see Supplementary Note 2)[28]. In this limiting case, the pressure profile at intermediate volumes is almost completely flat, as it takes little energy to create and propagate the lp/cp interface. This confirms the experimental observation that size-dependent flexibility is only encountered for smaller mesosized SPCs but not for larger microcrystals.

**Pressure-induced phase coexistence in DMOF-1(Zn).** As the pressures that stabilize the MIL-53(Al) coexistence regions are negative, a direct link with experimental observations cannot be established. To explore how the here identified phase coexistence could be reached experimentally, its occurrence in other SPCs is explored. To this end, the pillared-layer DMOF-1(Zn) is considered first[35]. At 300 K and ca. 200 MPa, DMOF-1(Zn) exhibits an lp-to-cp transition with a volume contraction of almost 50%, which was until recently assumed to occur cooperatively[21].

In Fig. 4, the 300 K pressure and free enthalpy profiles are depicted for a $8 \times 2 \times 8$ DMOF-1(Zn) mesocell. The free enthalpy, which follows immediately from our protocol (see Supplementary Note 6), is the appropriate thermodynamic potential to identify the (meta)stable states at constant temperature and pressure. Figure 4 reveals that also DMOF-1(Zn) mesocells exhibit $(n_{cp}, n_{lp})$ phase coexistence regions, which become metastable in the pressure range between 100 MPa and 140 MPa. While this phase coexistence resembles the observations in ref. [8], our methodology additionally reveals how well-chosen thermodynamic treatments could stabilize this phase coexistence. Such a hypothetical pressure treatment is proposed in Fig. 4d. When first increasing the pressure to just above the lp-to-cp transition pressure of 180 MPa, the lp state becomes unstable and the material starts its transition towards the cp state ($1 \rightarrow 2$). If the pressure is maintained (green treatment, $2 \rightarrow 3b$), this lp-to-cp transition is fully executed, thereby crossing the different phase coexistence regions. However, if the pressure is rapidly decreased to 140 MPa after reaching the transition pressure (blue treatment, $2 \rightarrow 3a$), DMOF-1(Zn) can be pressure quenched into the ($n_{cp} = 2, n_{lp} = 6$) metastable state at 140 MPa. According to these observations, phase coexistence in DMOF-1(Zn) could potentially be accessed experimentally by a proper pressure quenching treatment.

**Temperature-induced phase coexistence in MIL-53(Al)-F.** Experimentally, flexibility in SPCs was also observed under influence of other triggers such as temperature changes and guest adsorption. In this respect, it is important to investigate whether dedicated temperature and adsorption treatments can also instigate spatial disorder and phase coexistence in SPCs. Temperature-induced phase coexistence is here investigated for MIL-53(Al)-F, the isoreticular analog of MIL-53(Al) that is obtained by substituting the OH$^-$ framework anion with the larger F$^-$ anion[36]. This substitution stabilizes the lp state, as confirmed by our free energy profiles in Fig. 5a.

At 100 K and 0 MPa, layer-by-layer phase coexistence stabilizes MIL-53(Al)-F at intermediate volumes between about 1000 Å$^3$ and 1325 Å$^3$, leading to three metastable phase coexistence states at about 1125 Å$^3$, 1225 Å$^3$, and 1320 Å$^3$ (open circles). However, when increasing the temperature to 300 K, the intermediate phase coexistence regions and the volume region for which phase separation occurs are reduced. Increasing the temperature even more, to 500 K, leads to a complete disappearance of phase coexistence. This is assumed to originate from entropic effects, which were shown to destabilize the cp phase at higher temperatures[22], thereby also preventing phase coexistence. Rather, only dynamic phase mixing is observed, during which the framework continuously switches between a pure lp phase and different mixed phases. This dynamic disorder at higher temperatures is a consequence of the intermediate phase coexistence regions being less stable than the pure cp or lp phase. While the barriers between the coexistence regions and the pure phases are high enough to stabilize phase coexistence at low temperatures, they are more easily overcome at higher temperatures due to thermal fluctuations, leading to the disappearance of phase coexistence when increasing the temperature.

The temperature-dependent phase coexistence in MIL-53(Al)-F implies that also dedicated temperature treatments could stabilize phase coexistence. Such an experimental temperature treament is proposed in Fig. 5d. When heating a MIL-53(Al)-F sample in its cp phase from 100 K to 500 K ($1 \rightarrow 2$) and maintaining the temperature afterwards ($2 \rightarrow 3b$), a cp-to-lp transition is induced, as predicted computationally for the mesocell in Fig. 5 (green treatment). However, if one would quench the system at point 2, a metastable phase coexistence region could be accessed, corresponding to the blue treatment ($1 \rightarrow 2 \rightarrow 3a$) in Fig. 5d.

**Adsorption-induced phase coexistence in CoBDP.** The possibility to induce phase coexistence through guest adsorption is investigated for methane adsorption in CoBDP[37]. This material has been highlighted recently due to its promising performance for gas storage and its multi-step breathing behavior[37,38]. Figure 6 indicates the pressure and associated free energy profiles for CoBDP at 300 K with increasing methane content. The empty CoBDP mesocell displays intrinsic phase coexistence regions, which are metastable even at atmospheric pressure (open circles on the free energy profile). Upon increasing the methane content, however, these metastable regions start to disappear. As methane adsorption in CoBDP strongly destabilizes the cp phase, most $(n_{cp}, n_{lp})$ coexistence regions are destabilized upon methane adsorption. At sufficiently low methane adsorption, some coexistence regions remain metastable as the methane molecules may be located only in the lp layers, whereas the cp layers remain empty (see Supplementary Movie 11). This is similar to theoretical observations for methane adsorption in MIL-53(Sc)[39]. At higher methane content, however, this methane separation is no longer possible, and only a stable lp phase remains.

As phase coexistence in CoBDP depends on the amount of adsorbed molecules, methane adsorption could potentially be used to experimentally trigger phase coexistence in CoBDP. However, computationally identifying such an experimental treatment would require determining the osmotic potential to take into account methane adsorption and desorption, which is

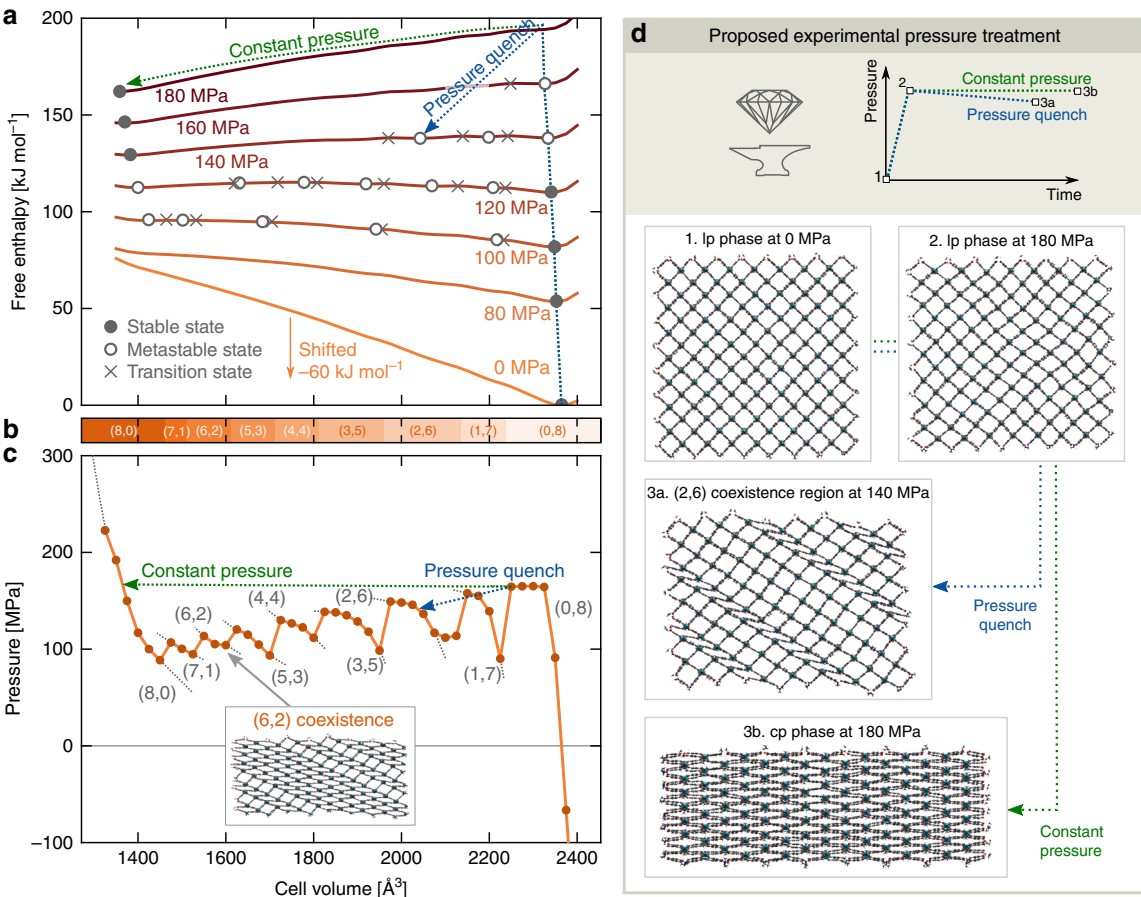

**Fig. 4** Using pressure as a stimulus to access the metastable cp/lp coexistence regions in DMOF-1(Zn). **a** Free enthalpy and **c** pressure equations of state, with **b** indication of the metastable $(n_{cp}, n_{lp})$ phase coexistence regions as a function of the volume for an $8 \times 2 \times 8$ mesocell of DMOF-1(Zn) at 300 K. The different (meta)stable states and transition states at different pressures are indicated on the free enthalpy equations of state with (open) circles and crosses, respectively. **d** The material's response upon two different pressure treatments is predicted: a constant-pressure treatment (green) and a pressure quenching treatment (blue), demonstrating how the latter leads to a stabilized (2,6) phase coexistence. The 0 MPa free enthalpy profile is globally shifted over +60 kJ mol$^{-1}$ to aid the visualization

still computationally too expensive for the system sizes considered here (see Methods) [21,40].

## Discussion

The results presented above illustrate that downsizing SPCs to mesosized crystals offers the potential to strongly alter their dynamic nature[24–28]. To connect these findings with the weak interactions dominating the anomalous behavior of SPCs, it is interesting to explore the analogy with a mechanic winerack structure. Macroscopically, phase transitions in rigid-linker SPCs are often simplified to the hinging of rigid organic spokes around the rigid joints of such a winerack structure[41]. If this hinging were to be frictionless, phase transformations to the energetically most favored conformation would occur collectively inside these materials. However, the lp/cp interfacial defects in Figs. 4–6 reveal that for realistic transitions, molecular deformations occur locally at the hinges between the two phases. As these deformations introduce friction during the transition, they suppress the cooperative transition mechanism. Experimental SPCs can therefore exhibit more localized phase transitions that imply phase coexistence and propagate through the crystal as a wavefront at a rate determined by the lattice vibrations associated with the molecular-level deformations at the hinges.

In this work, we investigated how crystal size and thermodynamic conditions affect this energetic barrier and the dynamic

behavior of pillared-layered SPCs. Their winerack structure leads to typical layer-by-layer phase coexistence which introduces cp/lp interfacial defects (see Fig. 1b) with an associated energetic barrier. As this barrier increases with decreasing crystallite size, phase coexistence in SPCs could lead to the experimentally observed suppression of the lp-to-cp phase transition in smaller crystals if this transition requires the instantaneous formation of interfacial defects[8,24–28]. As the observed phase coexistence remains metastable, phase coexistence within a given sample requires dedicated experimental treatments such as temperature or pressure quenching. Furthermore, due to the delicate balance between entropic and enthalpic contributions, high temperatures and high guest loadings were found to suppress phase coexistence. The thermodynamic criteria for phase coexistence identified in this work form a first guideline towards a fundamental understanding of how interfacial defects in SPCs form a type of spatial disorder that can be exploited to adjust their flexible nature, which we hope may form a stimulus to experimentally synthesize defect-engineered SPCs for functional applications.

## Methods

**Ab initio derived force fields**. To investigate the flexible nature of the four SPCs discussed in this manuscript, MIL-53(Al), DMOF-1(Zn), MIL-53(Al)-F, and CoBDP, material-specific force fields were used. These force fields were derived from ab initio input, using the QuickFF software—including cross terms—to describe the covalent interactions[42]. Furthermore, the electrostatics were modeled by Coulomb interactions between fixed Gaussian-like atomic charges. The

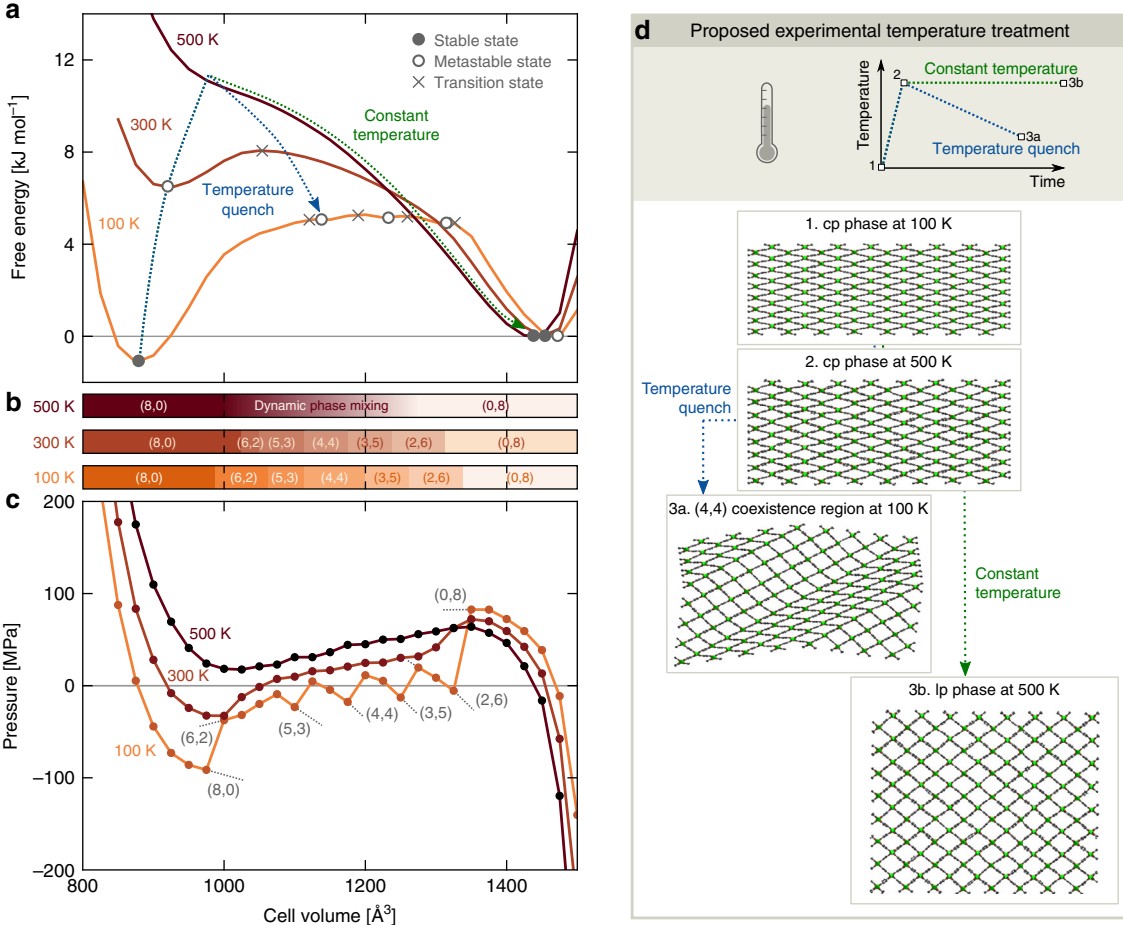

**Fig. 5** Using temperature as a stimulus to access the metastable cp/lp coexistence regions in MIL-53(Al)-F. **a** Free energy and **c** pressure equations of state at 100 K, 300 K, and 500 K, with **b** indication of the metastable $(n_{cp}, n_{lp})$ phase coexistence regions as a function of the volume for an $8 \times 2 \times 8$ mesocell of MIL-53(Al)-F. The different (meta)stable states and transition states are indicated on the free energy equations of state with (open) circles and crosses, respectively. **d** The material's response upon two different temperature treatments is predicted: a constant-temperature treatment (green) and a temperature quenching treatment (blue), demonstrating how the latter leads to a stabilized (4,4) phase coexistence

magnitude of these charges was determined through the Minimal Basis Iterative Stockholder partitioning scheme[43], whereas the Gaussian radii were computed according to the scheme of Chen and Martínez[44]. Finally, the van der Waals interactions were modeled according to the MM3 force field[45]. The obtained force fields were extensively compared with experimental data, ab initio data, and other force fields in ref. [21], especially focussing on an accurate description of the flexible nature of these materials.

**Thermodynamic ensembles**. The results reported in this work were obtained from simulations in either the $(N, P, \boldsymbol{\sigma}_a = \mathbf{0}, T)$ or the $(N, V, \boldsymbol{\sigma}_a = \mathbf{0}, T)$ ensemble, as introduced in ref. [33]. In the $(N, P, \boldsymbol{\sigma}_a = \mathbf{0}, T)$ ensemble, the number of particles $N$ (including the framework atoms and, if present, the adsorbed molecules), the mechanical pressure $P$, the deviatoric stress $\boldsymbol{\sigma}_a$, and the temperature $T$ are controlled. By controlling the total external stress $\boldsymbol{\sigma} = \boldsymbol{\sigma}_a + P\mathbf{1}$, both the volume and the shape of the simulation cell can fluctuate freely during the simulation. These $(N, P, \boldsymbol{\sigma}_a = \mathbf{0}, T)$ simulations were used to obtain the results of Fig. 2, as well as to generate initial structures for the results depicted in Figs. 3–6. In contrast, in the $(N, V, \boldsymbol{\sigma}_a = \mathbf{0}, T)$ ensemble, the unit cell volume $V$ is constrained rather than controlling the pressure. In this way, the simulation cell shape can still fluctuate freely, so that one can extract from this simulation the average pressure the material exerts on its environment at the given unit cell volume. As outlined below, this forms the basis for the contruction of the pressure equations of state. These $(N, V, \boldsymbol{\sigma}_a = \mathbf{0}, T)$ simulations were carried out on a predefined volume grid for each of the four SPCs to obtain the results of Figs. 3–6.

**Molecular dynamics simulations**. All simulations were carried out with LAMMPS for the efficient evaluation of the interatomic forces[46]. The LAMMPS engine was interfaced with our in-house software code Yaff[47]. To control the temperature during the simulation, a Nosé–Hoover chain thermostat of three beads and a

relaxation time of 0.1 ps was used[48–51]. Similarly, the pressure and/or deviatoric stress was controlled using a Martyna–Tobias–Tuckerman–Klein barostat with a relaxation time of 1 ps[52,53]. The equations of motion were updated through the Verlet scheme, using a time step of 0.5 fs to ensure energy conservation (0.75 fs for CoBDP). The electrostatic interactions were calculated using an Ewald summation with a real-space cutoff of 12 Å, a splitting parameter $\alpha$ of 0.213 Å$^{-1}$ and a reciprocal space cutoff of 0.32 Å$^{-1}$[54]. The van der Waals interactions were calculated with a smooth cutoff at 12 Å. The long-range interactions were supplemented by an analytical tail correction. The simulations of Figs. 3–6 were first equilibrated for 100 ps, followed by a 900 ps production run. As outlined in Supplementary Note 7, this ensures that the results are converged.

**Periodic boundary conditions**. Throughout this manuscript, periodic boundary conditions were assumed for each material. The adoption of periodic boundary conditions entails three major advantages over using large and isolated crystallites, even though they do not take into account possible surface effects as in ref. [8]. First off, when using periodic boundary conditions, the pressure applied on the system can be directly simulated using barostats, which dynamically modify the cell parameters during the simulation with the aim to control the pressure. When using isolated systems, this is no longer possible and other stimuli—which are typically more difficult to access and interpret experimentally—need to be used to force the material to breathe.

A second reason to adopt periodic boundary conditions is that, as noted above, the Ewald summation can be used to efficiently and accurately evaluate the electrostatic interactions in the system[54]. For large isolated systems, this is no longer possible, and approximate schemes need to be adopted which are typically less accurate as atoms may momentarily enter or exit the considered cutoff sphere. As these oscillations mainly occur in the volume region where one expects phase coexistence to take place, they can hamper the identification of possible metastable phase coexistence regions[8].

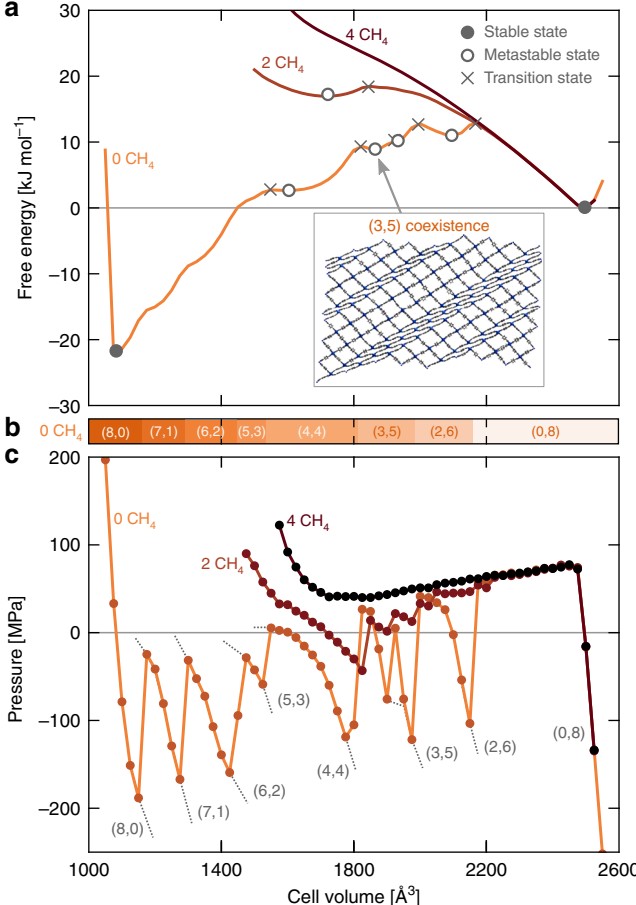

**Fig. 6** Using guest adsorption as a stimulus to access the metastable cp/lp coexistence regions in CoBDP. **a** Free energy and **c** pressure equations of state at loadings of 0, 2, and 4 methane molecules per conventional unit cell, with **b** indication of the metastable ($n_{cp}, n_{lp}$) phase coexistence regions as a function of the volume for the empty framework for an 8 × 2 × 8 mesocell of CoBDP at 300 K. The different (meta)stable states and transition states are indicated on the free energy equations of state with (open) circles and crosses, respectively

A third and final advantage of using periodic boundary conditions is that they facilitate the definition of the different subcells that form the material, as highlighted in Supplementary Note 3.

**Pressure, free energy, and free enthalpy equations of state**. In contrast to regular constant-temperature constant-pressure simulations, the here adopted pressure equations of state were specifically established to accurately sample the flexibility in SPCs under operating conditions of temperature, pressure, and guest loading[55]. To construct the pressure equation of state for a given material, a set of $(N, V, \sigma_a = 0, T)$ simulations are carried out in parallel at different volumes. The structures at these different volumes are obtained by preceding $(N, P, \sigma_a = 0, T)$ simulations at a pressure $P$ above the lp-to-cp transition pressure. These $(N, P, \sigma_a = 0, T)$ simulations thereby sweep over all intermediate states. Subsequently, structures are extracted according to a predefined volume grid and used as initial structures for the $(N, V, \sigma_a = 0, T)$ simulations. Following our earlier published protocol[33], we have taken care to perform the $(N, V, \sigma_a = 0, T)$ simulations at a pressure that is not too high, so to allow the material to relax during the transition. Additionally, the methane molecules were randomly inserted in CoBDP, requiring that the interaction energy was within reasonable limits (typically, $|E_{int}| < 20$ kJ mol$^{-1}$). As shown in Supplementary Note 8, the results reported here are largely insensitive to the initial structures used during the $(N, V, \sigma_a = 0, T)$ simulations, the exact location of the methane molecules in CoBDP, and the predefined volume grid.

At each volume, the $(N, V, \sigma_a = 0, T)$ simulation predicts the average pressure the material exerts on its environment. In equilibrium, this coincides

with the mechanical pressure that needs to be exerted on the material to retain the given volume. As a result, from this set of $(N, V, \sigma_a = 0, T)$ simulations, one can obtain the macroscopic $P(N, T; V)$ equation of state at a given temperature. By thermodynamic integration, the free energy $F(N, T; V) = -\int^V P(N, T; V')dV'$ can be accessed, whereas the free enthalpy follows as $G(N, T, P; V) = F(N, T; V) + PV$ (see also Supplementary Note 6 for a practical example for DMOF-1(Zn)).

As outlined in detail in ref. [55], the pressure equation of state directly reveals the different (meta)stable and transition states at a given temperature, pressure, and guest loading. To this end, it suffices to determine the intersections between the constructed pressure equation of state and a horizontal line, drawn at the pressure, either zero or nonzero, at which one wishes to obtain information. Any intersection with a positive $\partial P/\partial V$ slope is then a mechanically unstable state with a negative bulk modulus, whereas any intersection with a negative slope is a (meta)stable state at the given pressure. This information can also be obtained when considering the free energy (at constant volume and temperature) and the free enthalpy (at constant temperature and pressure) equations of state. Local minima in these thermodynamic potentials correspond to (meta)stable states at the given thermodynamic conditions, with the global minimum denoting the stable state, whereas maxima correspond to transition states. This protocol was adopted earlier to obtain microscopic insight in spatial disorder in rigid MOFs, such as the loss of crystallinity in UiO-66-like materials[17].

While flexibility triggered by temperature and pressure can be computationally predicted via the free energy and free enthalpy equations of state, respectively, these equations of state cannot be adopted to directly predict the experimental response of a material under gas adsorption and desorption. This is a consequence of simulating at a constant number of gas molecules in this protocol rather than at a constant chemical potential or gas pressure. To computationally predict flexibility under gas adsorption, one would rather need to construct the osmotic potential[56]. This osmotic potential can be accessed using hybrid schemes that contain both molecular dynamics and Monte Carlo steps—the latter accounting for the adsorption and desorption of guests, or analytically, by determining the Legendre transform of the free energy profiles[21,40]. In both cases, however, a far larger number of configurations need to be considered to obtained meaningful results, making the osmotic potential substantially more expensive to determine computationally.

**Thermodynamic model**. To understand how the different coexistence regions emerge in a given SPC as well as to extrapolate our results to both larger cells and other SPCs, a simple thermodynamic model was constructed. This thermodynamic model predicts the pressure equation of state and associated thermodynamic potentials based on equilibrium information of the two pure phases (relative stability, equilibrium volume, equilibrium bulk modulus and its derivatives) and the free energy associated with the formation of an interface between an lp and a cp layer. As outlined in more detail in Supplementary Note 1, the model first determines the free energy for each of the possible phase coexistence regions, assuming that these coexistence regions do not interact, and determines the corresponding pressure equation of state. Afterwards, the Gibbs rule is applied to interpolate between adjacent coexistence regions, giving rise to the red curves in Fig. 3. From Fig. 3, it is clear that this simple model can qualitatively predict for which volumes a given ($n_{cp}, n_{lp}$) phase coexistence region will emerge. To obtain quantitative correspondence with the pressure equation of state, a more elaborate model needs to be constructed.

## Data availability

Computational data supporting the results of this work are available from the online GitHub repository at https://github.com/SvenRogge/supporting-info or upon request from the authors.

## Code availability

The Yaff software used to perform the MD simulations in this manuscript is freely accessible via https://molmod.ugent.be/software/yaff. Representative input and processing scripts are available at https://github.com/SvenRogge/supporting-info.

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

## Acknowledgement

This work is supported by the Fund for Scientific Research Flanders (FWO) through a postdoctoral fellowship for S.M.J.R. (Grant no. 12T3519N) and the Research Board of Ghent University (BOF). Funding was also received from the European Union's Horizon 2020 Research and Innovation Programme (ERC Consolidator Grant Agreement 647755 — DYNPOR (2015–2020)). The computational resources and services used in this work were provided by the VSC (Flemish Supercomputer Center), funded by the Research Foundation - Flanders (FWO) and the Flemish Government – department EWI.

## Author contributions

S.M.J.R., M.W. and V.V.S. initiated the discussion, designed the paper, and were involved in the discussion of the results. S.M.J.R. and V.V.S. wrote the manuscript with contributions of all authors. S.M.J.R. performed all simulations.

## Competing interests

The authors declare no competing interests.
