## [Peer Review File · Nature Communications]

Reviewers' comments:

Reviewer #1 (Remarks to the Author):

Rogge et al. have presented a comprehensive investigation of the influence size effects have on phase changes in flexible porous crystalline processes, using molecular simulation. In particular, the investigation considers three metal-organic framework (MOF) structures, each with a wine-rack motif, under the influence of pressure, temperature and adsorption. This is high-quality work which builds on the authors previous studies on characterising the mechanical properties of soft porous crystals.

I believe the quality and potential influence of this work is commensurate with the quality expected by readers of Nature Communications and is probably publishable subject to a number of revisions as discussed below.

Importantly, the authors do not reference a study, which is also currently under review, shared on the ChemRxiv preprint server [10.26434/chemrxiv.8281082]. This work also investigates the effect of size on a flexible process in a wide-rack MOF, but in the absence of periodic boundary conditions. This work also highlights this layer-by-layer phase transition and the stabilisation of intermediate volumes. While I don't believe the work by Keupp et al. negates the novelty of the present work, it should be referenced and discussed.

The results presented in Fig 2 illustrate the size-dependent mechanism with respect to time, however, owing to the authors' previous work assessing barostats for this application [10.1021/acs.jctc.5b00748], I am unsure if this can be considered a fair comparison. The simulations were conducted with the same barostat relaxation time but with very different numbers of atoms, thus the time to transition is not necessarily comparable. As a result, the intermediate phases may be transient. Moreover, the noughts and crosses scheme used in this figure represents the phases in three discrete ways, however, the volume attributed to the intermediate or unstable phases spans small and large volumes. In a majority of the cases, the intermediate volumes appear more similar to a cp phase than an lp phase, thus a single state for these phases is misrepresentative.

A large proportion of the pressure-volume equation of states shows discontinuous pressure changes associated with the collapse of each layer. Have the authors considered smaller volume steps in an effort to capture the transition pressures during the layer collapses? Further, the choice of volume scan, be it the step size or the initial configurations plays an extremely important role in the formation of the coexistent phases. The manuscript does not state how the initial configurations for

each volume were produced, and I am unsure if the phases observed are independent of these initial configurations. Have sequential volume scans also been tested and produce the same results?

Given the stochastic process behind the formation of these phases and the disordered arrangement they may show, I am unsure if single trajectory simulations are able to establish these metastable phases. One of the key outcomes highlights hypothetical experimental treatments to obtain the proposed disordered phases. Can a single NPT trajectory, with the suggested pressure or temperature quenching, produce the proposed phases? If this is not the case, can it be considered possible by experiment?

Finally, the case study of adsorption-induced changes lacks the same scientific robustness as the other case studies and requires additional examination. For example, how were the configurations for the adsorbed molecules produced? There are many possibilities for the placement of methane molecules, and the equation of state, particularly the transition region, is sensitive to their configuration. Furthermore, the arrows and suggested treatment shown in Fig 6 are not physical. The free energy profiles are missing specific terms relating to adsorption and desorption. Although the figure plots the energy of each op phase at 0, this is not necessarily physical. Without having the adsorption terms, the energy difference between the curves are not comparable, so you can not move between these curves as depicted. This renders the adsorption-induced case study ill-defined, and the results do not have the same certainty as the other case studies.

Reviewer #2 (Remarks to the Author):

This manuscript reports a molecular simulation investigation complemented by a thermodynamic modeling of the size-dependency of the phase transition behavior in flexible MOFs often called after Kitagawa : Soft Porous Crystals (SPCs). In these crystals the transitions between the different forms of the flexible material ("narrow" or "large" forms for instance) have been assumed to take place cooperatively throughout the whole material. This is the case for instance in the model proposed by Triguero et al describing MIL-53 transition behavior (reference 31 in the manuscript, and also J. Chem. Phys. 137 (2012) 184702). In the present study, the authors investigate these phenomena in the mesoscale size regime using rather large simulation samples, and demonstrate that spontaneous spatial disorder may be generated during the transition process, giving rise to what is called "interfacial defects", a kind of grain boundary defect between two competing forms of the flexible crystal. This was not observed in previous works performed on conventional simulation nanocells with periodic boundary conditions, presumably because their size was too small to accommodate spatial disorder.

Several SPCs were investigated here ranging from Wine Rack to Pillared layers structures and in all cases it has been possible to provide an understanding of the thermodynamic conditions under which phase coexistence was possible. This opens the way to stabilize phase coexistence (by temperature or pressure quenching for instance) in mesocrystals, displaying new and interesting physicochemical properties.

This manuscript reports very important new results in the field of MOFs and SPCs crystalline properties. The interest in defective MOFs is growing in the international community. I recommend publication of this work in Nature Comm.

The authors may consider the following comment.

This work addresses the crystal size downsizing effect that have been observed in several experiments. If my understanding of the present work is correct, the results do not quite explain for the time being the suppression of the LP to NP transition in MIL-53, observed by Kitagawa I think. I though the authors might add a comment on this point.

Reviewer #3 (Remarks to the Author):

In this article, Rogge et al. present a computational investigation of the phase separation in MOFs, and how this is dependent upon the size of the cell used in the simulation. In particular, observing that you get phase separation if you have larger (“mesocell”) cells. The work is thoroughly presented and clearly written.

The use of soft porous crystals (SPCs) is overstating the breadth of what is covered, when only a series of MOFs are covered. So I think it would be better if they made use of the term MOF in the title instead.

I think the work would be of interest to the community although I think it would be less so to the wider field, given it is looking at a specific phenomenon in MOFs (or soft porous crystals).

Reviewer #1

Rogge *et al.* have presented a comprehensive investigation of the influence size effects have on phase changes in flexible porous crystalline processes, using molecular simulation. In particular, the investigation considers three metal-organic framework (MOF) structures, each with a wine-rack motif, under the influence of pressure, temperature and adsorption. This is high-quality work which builds on the authors previous studies on characterising the mechanical properties of soft porous crystals.

I believe the quality and potential influence of this work is commensurate with the quality expected by readers of *Nature Communications* and is probably publishable subject to a number of revisions as discussed below.

We would like to thank the reviewer for his/her careful reading of the manuscript and the positive evaluation. We have considered all comments and suggestions raised by the reviewer, which allowed us to improve the manuscript substantially. In what follows, detailed answers to all comments are given.

- 1. Importantly, the authors do not reference a study, which is also currently under review, shared on the ChemRxiv preprint server [10.26434/chemrxiv.8281082]. This work also investigates the effect of size on a flexible process in a wide-rack MOF, but in the absence of periodic boundary conditions. This work also highlights this layer-by-layer phase transition and the stabilisation of intermediate volumes. While I don't believe the work by Keupp *et al.* negates the novelty of the present work, it should be referenced and discussed.**

The work of Keupp *et al.* [1] has been performed completely independently of our submitted work, which make the conclusions of both studies even more interesting and challenging as they have been accomplished at the same time. Because of the almost simultaneous submission of the two papers, we were not able to cite that manuscript and vice versa at the time of submission. However, as the manuscript of Keupp *et al.* is now available to the community via the ChemRxiv preprint server [1], it has been taken up in the revised reference list. At the same time a comparative study of both works is provided both here and in the manuscript. This discussion gives an extra dimension to the contents of our paper.

Both the work of Keupp *et al.* and our work aim to understand the size-dependent phase transition in wine-rack-type metal-organic frameworks (MOFs), which we both find to occur in a layer-by-layer fashion. As the computational approaches followed by both groups differ substantially, the two works are complementary and demonstrate, from different points of view, that the assumption of collective flexibility in pillared-layered MOFs does not hold for more realistically sized MOF crystals. As both works reach similar conclusions using a different approach and using different models to describe the materials, they highlight the transferability and importance of the conclusions drawn in both manuscripts.

When comparing in more detail the approach followed in our manuscript to the one adopted in the work of Keupp *et al.* [1], three major differences can be distinguished, which are discussed in more detail hereunder.

Periodic boundary conditions. First of all, while we used periodic boundary conditions for our simulations, Keupp *et al.* constructed isolated nanocrystallites with different sizes [1]. As a result, they take into account surface effects that may facilitate the nucleation of the phase transition. From the results reported in Figure 7 and 9 of their preprint manuscript, it indeed seems that these surface regions do form preferential nucleation sites in their simulations. This is an effect that is not accounted for in our simulations, although it should be noted that the relative importance of these surface effects is expected to decrease the larger the considered cell as also the external-surface-to-volume ratio decreases.

Our choice to adopt periodic boundary conditions instead of considering isolated nanocrystals, despite the limitation of neglecting possible surface effects, is based on three observations. First off, a periodically repeated simulation cell allows one to directly apply a hydrostatic pressure to the material via so-called barostats [2]. These barostats dynamically modify the simulation cell matrix during the simulation with the aim to control the pressure. Using our approach relying on periodic simulations, the pressure can therefore be easily extracted, giving rise to the macroscopic pressure equations of state. Pressure has not only been demonstrated to be an important stimulus to induce flexibility in a large variety of soft porous crystals but, importantly, is also easily accessible from an experimental point of view using, for instance, mercury intrusion. As a result, our pressure equations of state can be directly validated experimentally which has proven crucial to confirm the accuracy of this protocol and, in the present case, is essential to experimentally access phase coexistence. Moreover, our protocol based on these pressure equations of state does not only allow for the determination of pressure-induced flexibility, but can also be adopted to study phase transitions that are induced by temperature changes or guest adsorption, as demonstrated with the case studies in this manuscript and in Ref. [3].

A second reason to adopt periodic boundary conditions is that the efficient Ewald-type calculations of the Coulombic interactions that are used in this manuscript are prohibitive for isolated systems, as was also noted by Keupp *et al.* [4]. As a result, approximate schemes need to be adopted to model isolated systems which are typically less accurate as atoms may momentarily enter or exit the considered cutoff sphere [1]. This results in oscillations in the pressure equations of state, which mainly occur in the volume region where one expects phase coexistence to take place [1]. These oscillations therefore hamper the identification of possible metastable phase coexistence regions [1].

A third and final advantage of using periodic boundary conditions is that they facilitate the definition of the different subcells that form the material. As we highlighted in the Supplementary Note 3, the definition of the subcells in periodic simulations remains straightforward, with only a small relative volume that is not accounted for during the simulation. For isolated systems, however, defining the subcells becomes substantially less straightforward [1].

Stabilization of phase coexistence. The scope of our manuscript is twofold. At first instance, we highlight that phase transitions in soft porous crystals do not necessarily occur collectively but rather in a layer-by-layer fashion that is affected by the size of the crystal. This implies dynamic spatial disorder in the material during the transition, resulting in interfacial defects and an associated barrier to form these defects. This observation is similar to what is observed by Keupp *et al.* [1]. Our second key message is that this phase coexistence can be stabilized at intermediate volumes, which allows us to put forward potential experimental treatments such as pressure and temperature quenching to experimen-

tally access the here identified phase coexistence regions. The latter is an important step towards controlling and exploiting phase coexistence in soft porous crystals for engineering applications.

Selection of the external stimuli and materials. Finally, in our manuscript, we adopted a varied set of four pillared-layered MOFs: MIL-53(Al), DMOF-1(Zn), MIL-53(Al)-F, and CoBDP. As these materials exhibit flexibility under different thermodynamic stimuli – pressure, temperature and gas adsorption – considering a broader set of materials allowed us to demonstrate that the observed phase coexistence is not a system-specific phenomenon, but is rather a generic effect present in a variety of soft porous crystals which can be induced by different easily accessible and easily interpretable thermodynamic stimuli. Furthermore, as these four materials are each described with different models, the occurrence of phase coexistence in all these materials further underpins the transferability of the conclusions drawn in our manuscript.

To properly address the preprint work, we have altered our manuscript at various instances. In the introduction (page 3), we explicitly refer to the work of Keupp *et al.*:

“[...] we reveal that multiple phases can coexist in various pillared-layered SPCs, giving rise to interfacial defects (see Figure 1(a,b), similar to very recent observations by Keupp et al. [1].”

In the description of our systems (page 4), we specifically highlight that we adopt periodic boundary conditions:

“To this end, the different cell sizes [...] were considered [...], assuming periodic boundary conditions (see Methods).”

In the Methods section (pages 17-18), we introduced a complete paragraph to rationalize why periodic boundary conditions were used and compare this to Ref. [1], largely following the discussion above:

“Throughout this manuscript, periodic boundary conditions were assumed for each material. The adoption of periodic boundary conditions entails three major advantages over using large and isolated crystallites, even though they do not take into account possible surface effects as in Ref. [1]. First off, when using periodic boundary conditions, the pressure applied on the system can be directly simulated using barostats, which dynamically modify the cell parameters during the simulation with the aim to control the pressure. When using isolated systems, this is no longer possible and other stimuli – which are typically more difficult to access and interpret experimentally – need to be used to force the material to breathe.

A second reason to adopt periodic boundary conditions is that, as noted above, the Ewald summation can be used to efficiently and accurately evaluate the electrostatic interactions in the system [4]. For large isolated systems, this is no longer possible, and approximate schemes need to be adopted which are typically less accurate as atoms may momentarily enter or exit the considered cutoff sphere [1]. As these oscillations mainly occur in the volume region where one expects phase coexistence to take place, they can hamper the identification of possible metastable phase coexistence regions [1].

A third and final advantage of using periodic boundary conditions is that they facilitate the definition of the different subcells that form the material, as highlighted in Supplementary Note 3.”

When first discussing the layer-by-layer transition mechanism (page 6), we also refer to the work of Keupp *et al.*:

*"[...] leading to the peculiar layer-by-layer breathing in Figure 2 that was postulated earlier by Triguero *et al.* [5] and which was also observed in Ref. [1] for DMOF-1(Zn)."*

When stating on page 9 that the activation barrier for the formation of interfacial defects decreases with increasing crystal size, we emphasize that this was also observed by Keupp *et al.*:

*"The model therefore confirms the experimental observation that the barrier to introduce an interface increases the smaller the crystal [6], a result also obtained independently by Keupp *et al.* [1]."*

When discussing the phase coexistence in DMOF-1(Zn) on page 11, we compare to the results obtained by Keupp *et al.*:

"While this phase coexistence resembles the observations in Ref. [1], our methodology additionally reveals how well-chosen thermodynamic treatments could stabilize this phase coexistence."

Finally, when discussing the correlation of neighboring subcells along the inorganic chain direction in Supplementary Note 4 (page S-11), we also refer to the work of Keupp *et al.*:

*"This 1D correlated behavior was also observed, for instance, in the very recent work of Keupp *et al.* in which the thermal opening of DMOF-1(Zn) was studied for various crystallite sizes [1]."*

- 2. The results presented in Fig 2 illustrate the size-dependent mechanism with respect to time, however, owing to the authors' previous work assessing barostats for this application [10.1021/acs.jctc.5b00748], I am unsure if this can be considered a fair comparison. The simulations were conducted with the same barostat relaxation time but with very different numbers of atoms, thus the time to transition is not necessarily comparable. As a result, the intermediate phases may be transient. Moreover, the noughts and crosses scheme used in this figure represents the phases in three discrete ways, however, the volume attributed to the intermediate or unstable phases spans small and large volumes. In a majority of the cases, the intermediate volumes appear more similar to a cp phase than an lp phase, thus a single state for these phases is misrepresentative.**

The spatial disorder observed in the mesocell of Figure 2(b) is indeed a transient and dynamic effect. As we observed earlier for the nanocell of Figure 2(a), the dynamics of the lp-to-cp phase transition, including the time needed for the barostat to induce the transition, is strongly affected by the choice of barostat relaxation time [2]. Furthermore, the dynamics of the transition is also influenced by the system size, as the relative fluctuations in the instantaneous pressure decrease with the square root of the number of atoms [7]. To illustrate this, Figure 1 demonstrates how the system size impacts the dynamics of the lp-to-cp transition for different MIL-53(Al) simulation cells, showing that even the occurrence of a phase transition at a given pressure in these $(N, P, \sigma_\alpha = \mathbf{0}, T)$ simulations is size dependent for pressures just below the transition pressure. We therefore completely agree with the statement that the time to transition between the nanocell and the mesocell is not strictly comparable.

Figure 1. Evolution of the volume of the different subcells (colored lines) and the volume of the total simulation cell normalized on the number of subcells (dark gray line) as a function of the simulation time during an $(N, P, \sigma_a = \mathbf{0}, T)$ simulation at 300 K for a $1 \times 2 \times 1$ nanocell, a $2 \times 2 \times 2$ nanocell, and a $8 \times 2 \times 8$ mesocell of MIL-53(Al), starting in the large-pore phase. The pressures correspond, from top to bottom, to 0 MPa, 20 MPa (both below the transition pressure), 40 MPa (close to the transition pressure), and 60 MPa (above the transition pressure).

In Figure 2 of the manuscript, however, our aim is not to compare the time to transition, but rather to qualitatively compare the transition mechanisms present in the nanocell and the mesocell. To point towards this difference in transition time while simultaneously emphasizing that our focus lies on comparing the transition mechanism, we have altered the statement on page 6 to read:

“For the mesocell, the transition time differs due to the larger structure, but, more importantly, the transition itself also propagates differently through the material.”

Furthermore, we have added a discussion on the system size dependence of the dynamics of the transition during an $(N, P, \sigma_a = \mathbf{0}, T)$ simulation, including Figure 1 of this reply, to the Supplementary Note 5. In the discussion, we also clearly contrast this observed size dependence of the dynamics during $(N, P, \sigma_a = \mathbf{0}, T)$ simulations with the observation that the transition pressures extracted from our pressure equations of state reported in the manuscript are, in contrast, independent of the cell size.

Our goal of unraveling the transition mechanism rather than the time to transition is furthermore emphasized throughout the paragraph, with the main conclusion that the collective nature of the flexibility in the nanocell of Figure 2(a) is an artifact arising from considering too small a simulation cell, and is no longer present in the mesocell of Figure 2(b). As we fully acknowledge the transient nature of this transition, we have taken the utmost care in the original manuscript not to use the phrasing “phase coexistence” when describing Figure 2, but rather used the term “spatial disorder”. Phase coexistence would require that the two phases coexisting in the material are both (meta)stable, which is shown to occur later in the manuscript. In Figure 2, however, the closed-pore and large-pore phases never coexist; only cells in the unstable volume region (which is not a proper phase) coexist with cells in either the closed-pore or the large-pore phase.

To even further emphasize the dynamic and transient nature of the spatial disorder observed in Figure 2, we have further specified the statement on page 6:

“Correspondingly, the lp-to-cp transition spontaneously introduces transient and dynamic spatial disorder in the mesocell of Figure 2.”

Finally, we thank the reviewer for his/her comment on the symbols used to represent the different phases in Figure 2 of the manuscript. We would like to emphasize that two distinct classifications were used to characterize the different subcells in this figure. The first classification, using the symbols, is a mechanical one. This classification is uniquely defined based on the pressure equation of state for this material. As indicated in Figure 3 of the manuscript, this equation of state exhibits three regions as a function of the volume for small MIL-53(Al) cells: two mechanically stable regions with a positive bulk modulus, corresponding to the closed-pore (cp, circles in Figure 2) and large-pore (lp, squares in Figure 2) phases, and an intermediate unstable region with a negative bulk modulus (crosses in Figure 2). Subcells in the unstable volume region therefore differ critically from both the closed-pore or large-pore subcells as they are mechanically unstable, a distinction that is not immediately clear when only depicting the volume. The second classification, using the color code, simply considers the subcell volume as a structural and continuous designation.

As both classifications are complementary and superimposed here, we would suggest to keep both in Figure 2 to consider both the mechanical stability and the volume of the subcells. However, we empha-

sized the distinction between the two classifications in the caption of Figure 2 more strongly to avoid any confusion. The relevant part of this caption now reads:

"[...] The atomic visualizations are supplemented by schematic representations that are color coded based on the volume of the different subcells (structural classification). Furthermore, the mechanically stable closed-pore (cp) and large-pore (lp) phases and the intermediate unstable region are indicated by circles, squares, and crosses, respectively (mechanical classification). [...]"

The mechanical classification is also emphasized in the last paragraph on page 4:

"The subcells (see Supplementary Note 3) are marked with squares and circles to indicate the (meta)stable lp and cp phase, respectively, whereas crosses indicate the mechanically unstable volume region that separates both phases."

- 3. A large proportion of the pressure-volume equation of states shows discontinuous pressure changes associated with the collapse of each layer. Have the authors considered smaller volume steps in an effort to capture the transition pressures during the layer collapses? Further, the choice of volume scan, be it the step size or the initial configurations plays an extremely important role in the formation of the coexistent phases. The manuscript does not state how the initial configurations for each volume were produced, and I am unsure if the phases observed are independent of these initial configurations. Have sequential volume scans also been tested and produce the same results?**

To rule out any appreciable effect of the volume grid or of the choice of initial configurations on the reported pressure-versus-volume equations of state, we have performed a substantial amount of new mesocell simulations for MIL-53(Al) and CoBDP (the latter being covered in our answer to the fifth remark raised by the reviewer), and have thoroughly analyzed the results. For the 8x2x8 supercell of MIL-53(Al), three different sets of initial configurations were considered:

- i. Set 1: The initial configurations on the initial volume grid (volume step of 25 \AA^3);
- ii. Set 2: New configurations, extracted from an independent $(N, P, \sigma_\alpha = \mathbf{0}, T)$ simulation, on the initial volume grid (volume step of 25 \AA^3);
- iii. Set 3: New configurations on a new and finer volume grid (volume step of 10 \AA^3).

The three resulting pressure equations of state, obtained from 500 ps $(N, V, \sigma_\alpha = \mathbf{0}, T)$ simulations at 300 K, are shown in Figure 2 (full lines). These three pressure equations of state are almost indistinguishable, safe from the interpolations in the coarser volume grids, and typically fall within a few MPa of each other. Importantly, also the finer volume grid shows the discontinuous pressure changes associated with the collapse of each layer.

In addition to these three new pressure equations of state, Figure 2 also includes the original pressure equation of state (dotted line). This original equation of state was also obtained with the initial structures of set 1. However, the exclusion rules for the noncovalent interactions in our Yaff+LAMMPS interface were chosen differently for the new sets of simulations than for the old set, effectively resulting in a slight alteration of the noncovalent interactions. As a result, comparing the new and old pressure equations of state for set 1 also allows one to quantify the sensitivity of our results on the description of the noncovalent terms. From Figure 2, one observes that the difference between these two equa-

tions of state is not negligible near lower volumes, where the noncovalent interactions are more important. However, despite this, both equations of state predict the same qualitative phase coexistence behavior, confirming the robustness of our predictions.

This robustness analysis has been taken up in the new Supplementary Note 8 and is referred to in the Methods section (p. 19). Moreover, in the same paragraph, we have further specified the procedure to extract the initial configurations for each volume:

Figure 2. Pressure equations of state for a $8 \times 2 \times 8$ unit cell of MIL-53(Al) at 300 K and using different volume grids and/or initial structures. The full lines represent the structures obtained using three sets of initial structures that were chosen independently from each other. The blue and green structures are independently obtained along a volume grid with a volume spacing of 25 \AA^3 , whereas the orange structures are obtained along a volume grid with a reduced volume spacing of 10 \AA^3 . For these three sets of simulations, results at intermediate simulation times (100 ps, 200 ps, 300 ps, and 400 ps) are shown using lighter shades to demonstrate convergence has been reached. The dotted blue line represents the original 1 ns results reported in Figure 3(d) of the manuscript, starting from the initial structures of set 1 but with a different description of the noncovalent interactions. For all simulations a 100 ps equilibration time has been taken into account.

“The structures at these different volumes are obtained by preceding $(N, P, \sigma_\alpha = \mathbf{0}, T)$ simulations at a pressure P above the lp-to-cp transition pressure. These $(N, P, \sigma_\alpha = \mathbf{0}, T)$ simulations thereby sweep over all intermediate structures. Subsequently, structures are extracted according to a predefined volume grid and used as initial structures for the $(N, V, \sigma_\alpha = \mathbf{0}, T)$ simulations. Following our earlier published protocol [2], we have taken care to perform the $(N, P, \sigma_\alpha = \mathbf{0}, T)$ simulations at a pressure that is not too high, so to allow the material to relax during the transition. [...] As shown in Supplementary Note 8, the results reported here are largely insensitive to the initial structures used during the $(N, V, \sigma_\alpha = \mathbf{0}, T)$ simulations, the exact location of the methane molecules in CoBDP, and the predefined volume grid.”

- 4. Given the stochastic process behind the formation of these phases and the disordered arrangement they may show, I am unsure if single trajectory simulations are able to establish these metastable phases. One of the key outcomes highlights hypothetical experimental treatments to obtain the proposed disordered phases. Can a single NPT trajectory, with the suggested pressure or temperature quenching, produce the proposed phases? If this is not the case, can it be considered possible by experiment?**

To verify whether it is possible to directly access the proposed phase coexistence phases using $(N, P, \sigma_\alpha = \mathbf{0}, T)$ simulations mimicking the suggested pressure or temperature quenching treatments, we have carried out the suggested simulations for the 8x2x8 mesocells of DMOF-1(Zn) and MIL-53(Al)-F, and proposed a similar treatment for MIL-53(Al). The simulated treatments are visualized in the bottom panes of Figure 3 and compared to the corresponding experimental treatments put forward in the manuscript (top panes). Starting from the equilibrated large-pore (lp) and closed-pore (cp) phase of DMOF-1(Zn) and MIL-53(Al)-F (point 1 in Figure 3), we first suddenly increased the target pressure or temperature to induce a transition (point 2 in Figure 3), and afterwards suddenly decreased the target pressure or temperature according to the treatments proposed in the manuscript (point 3 in Figure 3). In addition, for MIL-53(Al), we started a 0 MPa $(N, P, \sigma_\alpha = \mathbf{0}, T)$ simulation in the equilibrated large pore (point 1 in Figure 3), suddenly increased the barostat target pressure to 45 MPa (point 2 in Figure 3), and then instantaneously changed the barostat target pressure to -100 MPa once the material starts its lp-to-cp transition (point 3 in Figure 3). In all three cases, we only varied the moment at which we started to applying the temperature or pressure quenching (the time of point 2).

Following these treatments, all materials show clear spatial disorder during the quenching process. However, phase coexistence regions could be identified only for the DMOF-1(Zn) mesocell. The two simulations for which phase coexistence regions could be identified, which differ only by the time of point 2 in Figure 3, are both visualized at https://www.dropbox.com/sh/s8i3mkzc4g1944o/AACKW-U6_ruW-C8V9m4QznBZa?dl=0. Furthermore, Figure 4 shows the evolution of the DMOF-1(Zn) mesocell for the quenching simulation in which (6,2) phase coexistence was observed. These simulations demonstrate the emergence of (6,2) and (7,1) phase coexistence regions in the material at point 3, which is located 5 ps after pressure quenching to 140 MPa. However, as indicated in Figure 4, when further extending these simulations to 10 ps, these phase coexistence regions eventually disappear in favor of a spatially disordered cp phase.

Given these observations, one could wonder (i) why phase coexistence regions can be permanently accessed in $(N, V, \sigma_a = \mathbf{0}, T)$ simulations, as consistently demonstrated in the manuscript, but not directly in $(N, P, \sigma_a = \mathbf{0}, T)$ quenching simulations, and (ii) what the repercussions of these observations are for experimental quenching experiments.

To answer point (i), we highlight the reviewer’s remark that, during the direct $(N, P, \sigma_a = \mathbf{0}, T)$ quenching simulations, whether or not to access a phase coexistence region is a stochastic effect. This is a well-known observation and is not specific to the phase coexistence observed here. To illustrate this, Figure 5 shows the free enthalpy equations of state for the $8 \times 2 \times 8$ mesocell of MIL-53(Al) at 300 K and pressures between -240 MPa and 80 MPa. At 40 MPa, this material still exhibits a metastable lp state, as indicated with the empty circle. However, when performing an $(N, P, \sigma_a = \mathbf{0}, T)$ simulation at 300 K and 40 MPa, we showed in Figure 1 that the material escapes from this metastable lp state, in favor of the stable cp state, due to instantaneous fluctuations in the internal pressure. Referring to Figure 5, this shows that instantaneous pressure fluctuations during $(N, P, \sigma_a = \mathbf{0}, T)$ simulations may steer a material over small free enthalpy barriers (“small” being a relative term determined by the size of the pressure fluctuations, which decrease inversely with the square root of the number of atoms in the system [7]). This effect was illustrated before in Ref. [2] and demonstrates that the occurrence of a permanent MIL-53(Al) lp phase in an $(N, P, \sigma_a = \mathbf{0}, T)$ simulation at 40 MPa and 300 K is a stochastic effect, even though the phase is metastable under these conditions.

Figure 3. Experimental treatments to access phase coexistence put forward in the manuscript (top) and the corresponding treatments used here to directly access phase coexistence using $(N, P, \sigma_a = \mathbf{0}, T)$ simulations (bottom). The square symbols on these treatments are reproduced from the manuscript, and correspond to (1) the equilibrated initial structure, (2) the initial structure brought to instability, and (3) the structure obtained after quenching with possible emergence of phase coexistence.

Similarly, for DMOF-1(Zn), the occurrence of a phase coexistence region during a direct $(N, P, \sigma_a = \mathbf{0}, T)$ simulation at thermodynamic conditions at which the free enthalpy equation of state indicates that this region is a metastable state, is a stochastic effect. From Figure 4 in the manuscript, it is clear that these phase coexistence regions form only shallow minima in the free enthalpy profile. As a result, instantaneous pressure fluctuations may indeed steer the material out of phase coexistence region, explaining the results shown in Figure 4. This also explains why our $(N, P, \sigma_a = \mathbf{0}, T)$ simulations revealed temporary phase coexistence in DMOF-1(Zn) but not in the other materials, as the DMOF-1(Zn) phase coexistence regions are more stable than those of MIL-53(Al) and MIL-53(Al)-F under the stated thermodynamic conditions. Furthermore, it explains why no permanent phase coexistence was observed in the work of Keupp *et al.* [1]. The observed stochastic nature of the $(N, P, \sigma_a = \mathbf{0}, T)$ simulations is exactly why we proposed in Ref. [2] to rather consider a set of $(N, V, \sigma_a = \mathbf{0}, T)$ simulations to characterize a material's response to an external pressure, as the $(N, V, \sigma_a = \mathbf{0}, T)$ ensemble avoids the system escaping from these metastable states. This approach was observed to reliably characterize the mechanical behavior of both rigid and flexible MOFs [8] and was therefore also adopted in this manuscript.

Figure 4. Evolution of the 8x2x8 DMOF-1(Zn) mesocell during the simulated $(N, P, \sigma_a = \mathbf{0}, T)$ quenching treatment depicted in Figure 3 showing (6,2) phase coexistence at point 3.

Although direct $(N, P, \sigma_\alpha = \mathbf{0}, T)$ quenching simulations cannot induce permanent phase coexistence, we are convinced that pressure and temperature quenching experiments would be able to access these phase coexistence regions. First off, many of the parameters during the $(N, P, \sigma_\alpha = \mathbf{0}, T)$ simulations (thermostat and barostat relaxation time, the speed with which the quenching takes place, the point at which the quenching starts...) profoundly affect the dynamics of the system [2, 9]. While Figure 3 shows that these parameters are chosen to closely mimic the experimental parameters, it is clear that a one-on-one correspondence cannot be reached – for instance, it would be impossible to obtain an instantaneous drop in pressure during the experiment, while this was perfectly possible in our $(N, P, \sigma_\alpha = \mathbf{0}, T)$ simulations. The most important difference between the proposed experiments and the $(N, P, \sigma_\alpha = \mathbf{0}, T)$ simulations performed here, however, is the system size. While we have enlarged our simulation cells beyond 10 nm in cell length, these are still about an order of magnitude smaller than experimental mesocells (see Figure 1(c) in the manuscript and Supplementary Note 2 for an overview). As outlined above, the instantaneous temperature and pressure fluctuations decrease inversely with the square root of the number of atoms in a system. As a result, these fluctuations will be substantially smaller in experimental crystals, thereby substantially decreasing the probability of escaping from the metastable phase coexistence regions and thus increasing the probability of inducing permanent phase coexistence in experimental crystals.

In conclusion, we are confident that the absence of permanent phase coexistence in direct $(N, P, \sigma_\alpha = \mathbf{0}, T)$ quenching simulations is a limitation of the choice of ensemble rather than a material property, which does not influence the experimental chances of permanently accessing phase coexistence.

Figure 5. Free enthalpy equations of state for the 8x2x8 mesocell of MIL-53(Al) at 300 K and for pressures between -240 MPa and 80 MPa. For each equation of state, the stable, metastable, and transition states are indicated by filled circles, empty circles, and crosses, respectively.

5. **Finally, the case study of adsorption-induced changes lacks the same scientific robustness as the other case studies and requires additional examination. For example, how were the configurations for the adsorbed molecules produced? There are many possibilities for the placement of methane molecules, and the equation of state, particularly the transition region, is sensitive to their configuration. Furthermore, the arrows and suggested treatment shown in Fig 6 are not physical. The free energy profiles are missing specific terms relating to adsorption and desorption. Although the figure plots the energy of each op phase at 0, this is not necessarily physical. Without having the adsorption terms, the energy difference between the curves are not comparable, so you cannot move between these curves as depicted. This renders the adsorption-induced case study ill-defined, and the results do not have the same certainty as the other case studies.**

The initial configurations of methane inside CoBDP were reproduced from Ref. [3]. Therein, the initial configurations were obtained by considering a trial insertion move of methane inside CoBDP, and accepting this move whenever the interaction energy was in between -20 kJ/mol and 20 kJ/mol. Different limits were tried, but gave very similar results. To check whether these initial positions substantially affect the obtained equations of state, we performed two new $(N, P, \sigma_\alpha = \mathbf{0}, T)$ simulations (one at 2 CH₄ and one at 4 CH₄ molecules per unit cell), during which the methane molecules were observed to diffuse freely. From both new $(N, P, \sigma_\alpha = \mathbf{0}, T)$ simulations, initial configurations were extracted according to the same volume grid as reported earlier. As a result, we obtained both for 2 CH₄ and 4 CH₄ molecules per conventional CoBDP unit cell a second and independent set of initial structures. These sets were then used to construct the pressure equations of state and compared to the pressure equations of state obtained with the original, independent set of structures.

The new pressure equations of state are reported as solid lines in Figure 6. As for MIL-53(Al) discussed in our reply to remark 3, the obtained equations of state and the observed phase coexistence is largely independent of the choice of initial conditions, demonstrating the robustness of our approach. The larger sensitivity is again obtained when altering the noncovalent force field terms (dotted lines).

In the Methods section (page 19), we refer to the robustness analysis in Supplementary Note 8 and have detailed our procedure to extract the initial structures:

“Additionally, the methane molecules were randomly inserted in CoBDP, requiring that the interaction energy was within reasonable limits (typically, $|E_{\text{int}}| < 20$ kJ/mol). As shown in Supplementary Note 8, the results reported here are largely insensitive to the initial structures used during the $(N, V, \sigma_\alpha = \mathbf{0}, T)$ simulations, the exact location of the methane molecules in CoBDP, and the predefined volume grid.”

In our original discussion of the adsorption-induced phase coexistence in CoBDP, we constructed the free energy equations of state at constant methane loading for different loadings. While this accurately reveals the different stable and metastable states at a given loading, including revealing the possibility of phase coexistence at intermediate volumes, the referee is correct that one would need to explicitly take into account the adsorption and desorption energy terms to directly propose an experimental treatment similar to the pressure- and temperature-induced phase coexistence. To this end, the osmotic potential should be determined at different chemical potentials or gas pressures [10], using for instance the hybrid methodology we proposed in Ref. [11] or via a Legendre transform of the free energy equations of state, as carried out for both MIL-53(Al) and DUT-49(Cu) in the Supplementary Information of Ref. [3]. In the hybrid methodology, the adsorption and desorption steps are taken into account by Monte Carlo steps, which are alternated with molecular dynamics simulations at constant guest load-

ing. As a result, this hybrid methodology effectively expands the phase space by having a variable number of guest molecules. Similarly, when considering the Legendre transform of the free energy equations of state, one would need to construct multiple equations of state at different guest loadings to determine the osmotic potential at a given chemical potential. When performing this Legendre transform in Ref. [3], for instance, 11 free energy equations of state were needed to obtain an accurate osmotic potential for xenon in a MIL-53(Al) nanocell, while 6 free energy equations of state were found to be insufficient to predict negative gas adsorption in DUT-49(Cu).

Figure 6. Pressure equations of state for a $8 \times 2 \times 8$ unit cell of CoBDP at 300 K and either two or four methane molecules per conventional unit cell, using two independent sets of initial structures. The dotted blue lines represent the original results reported in Figure 6 of the main manuscript. The full blue lines represent the new results obtained using the same initial structures (set 1). The orange lines represent the structures obtained using a new set of initial structures, chosen independently from the original initial structures (set 2). For the new simulations, results at intermediate simulation times (100 ps, 200 ps, 300 ps, and 400 ps) are shown using lighter shades to indicate convergence has been reached. The dotted lines represent the original 1 ns results reported in Figure 6 of the manuscript, starting from the initial structures of set 1 but with a different description of the noncovalent interactions. For all simulations a 100 ps equilibration time has been taken into account.

From these case studies, it is clear that determining the osmotic potential for the 8x2x8 CoBDP simulation cell (15.1 nm x 1.4 nm x 15.1 nm) would be a computationally very expensive challenge, requiring tens of free energy equations of state to perform the Legendre transform and obtain accurate results. Therefore, we report in the revised version of Figure 6 in the manuscript the occurrence of phase coexistence in this material at a given loading—which is independent of the adsorption/desorption terms, but omit the proposed experimental treatment. In addition, the last sentence of the introduction (p. 3) now reads:

“Based on these insights, we here hypothesize different pathways to experimentally observe phase coexistence in SPCs, paving the way to leverage spatially disordered SPCs for targeted applications.”

Furthermore, the last paragraph of the discussion on CoBDP (pages 14-15) is rewritten to read:

“As phase coexistence in CoBDP depends on the amount of adsorbed molecules, methane adsorption could potentially be used to experimentally trigger phase coexistence in CoBDP. However, computationally identifying such an experimental treatment would require determining the osmotic potential to take into account methane adsorption and desorption, which is still computationally too expensive for the system sizes considered here (see Methods) [3, 11].”

In the Methods section, when discussing the different relevant thermodynamic potentials (page 20), this is explained in more detail:

“While flexibility triggered by temperature and pressure can be computationally predicted via the free energy and free enthalpy equations of state, respectively, these equations of state cannot be adopted to directly predict the experimental response of a material under gas adsorption and desorption. This is a consequence of simulating at a constant number of gas molecules in this protocol rather than at a constant chemical potential or gas pressure. To computationally predict flexibility under gas adsorption, one would rather need to construct the osmotic potential [10]. This osmotic potential can be accessed using hybrid schemes that contain both molecular dynamics and Monte Carlo steps—the latter accounting for the adsorption and desorption of guests, or analytically, by determining the Legendre transform of the free energy profiles [3, 11]. In both cases, however, a far larger number of configurations need to be considered to obtain meaningful results, making the osmotic potential substantially more expensive to determine computationally.”

Reviewer #2

This manuscript reports a molecular simulation investigation complemented by a thermodynamic modeling of the size-dependency of the phase transition behavior in flexible MOFs often called after Kitagawa: Soft Porous Crystals (SPCs). In these crystals the transitions between the different forms of the flexible material ("narrow" or "large" forms for instance) have been assumed to take place cooperatively throughout the whole material. This is the case for instance in the model proposed by Triguero et al describing MIL-53 transition behavior (reference 31 in the manuscript, and also J. Chem. Phys. 137 (2012) 184702). In the present study, the authors investigate these phenomena in the mesoscale size regime using rather large simulation samples, and demonstrate that spontaneous spatial disorder may be generated during the transition process, giving rise to what is called "interfacial defects", a kind of grain boundary defect between two competing forms of the flexible crystal. This was not observed in previous works performed on conventional simulation nanocells with periodic boundary conditions, presumably because their size was too small to accommodate spatial disorder.

Several SPCs were investigated here ranging from Wine Rack to Pillared layers structures and in all cases it has been possible to provide an understanding of the thermodynamic conditions under which phase coexistence was possible. This opens the way to stabilize phase coexistence (by temperature or pressure quenching for instance) in mesocrystals, displaying new and interesting physicochemical properties.

This manuscript reports very important new results in the field of MOFs and SPCs crystalline properties. The interest in defective MOFs is growing in the international community. I recommend publication of this work in *Nature Comm*.

We would like to thank the reviewer for his/her careful reading of the manuscript and the positive evaluation. We have considered the comment raised by the reviewer, and improved the manuscript accordingly. In what follows, a detailed answer to this comment is given.

The authors may consider the following comment.

This work addresses the crystal size downsizing effect that have been observed in several experiments. If my understanding of the present work is correct, the results do not quite explain for the time being the suppression of the LP to NP transition in MIL-53, observed by Kitagawa I think. I though the authors might add a comment on this point.

We are not aware of any papers specifically discussing the suppression of the lp-to-cp transition in MIL-53(Al) upon crystal downsizing. However, multiple examples of the effect of crystal downsizing on the breathing behavior exists for other related soft porous crystals, such as $[\text{Cu}_2(\text{bdc})_2(\text{bpy})]_n$ [6], discussed by Kitagawa and co-workers, and MIL-53(Al)-NH₂, by Zhao and co-workers [12]. These examples are mentioned, among others, in the introduction of the manuscript, and it is very likely similar effects hold in MIL-53. The observed size-dependent suppression could have multiple origins, varying from the different amount of defects in crystals of different size [6], the larger surface-to-volume ratio of smaller crystals [1], or the increase in activation barrier to introduce phase coexistence upon decreasing crystallite sizes, as discussed here. For the moment, it remains unclear which of these hypotheses correspond best with the experimental observations.

To see how the hypothesis raised here, the size-dependent activation barrier to form lp/cp interfacial defects, may lead to the experimentally observed size-dependent suppression of the breathing behavior in soft porous crystals, consider two hypothetical crystals with a largely different crystallite size. For both crystals, it can be expected that the lp-to-cp transition, if it were to take place, would introduce spatial disorder and even phase coexistence in the system during the transition, as shown in Figure 2 of the manuscript. To form these phase coexistence regions, however, it is essential to create spatial disorder at the interfaces between the lp and cp phases. These interfacial defects can therefore only be formed if the system can overcome the activation barrier for the formation of these defects. As we demonstrated that this activation barrier scales inversely with the square root of the number of layers in the system, it is easier to introduce spatial disorder in the largest of the two hypothetical MIL-53 crystals. This would explain why it is more difficult to induce an lp-to-cp phase transition in the smallest of the two crystals; up to a point that the phase transition is completely suppressed once the crystal size drops below a given critical crystallite size.

To highlight this hypothesis, we have rewritten the last paragraph of the discussion section (p. 15). It now reads:

“In this work, we investigated how crystal size and thermodynamic conditions affect this energetic barrier and the dynamic behavior of pillared-layered SPCs. Their winerack structure leads to typical layer-by-layer phase coexistence which introduces cp/lp interfacial defects (see Figure 1(b)) with an associated energetic barrier. As this barrier increases with decreasing crystallite size, phase coexistence in SPCs could lead to the experimentally observed suppression of the lp-to-cp phase transition in smaller crystals if this transition requires the instantaneous formation of interfacial defects [1, 6, 12, 13, 14, 15].”

Reviewer #3

In this article, Rogge *et al.* present a computational investigation of the phase separation in MOFs, and how this is dependent upon the size of the cell used in the simulation. In particular, observing that you get phase separation if you have larger (“mesocell”) cells. The work is thoroughly presented and clearly written.

We would like to thank the reviewer for his/her careful reading of the manuscript and the very positive evaluation. We have considered the comment raised by the reviewer, and provided a detailed answer to this comment in what follows.

The use of soft porous crystals (SPCs) is overstating the breadth of what is covered, when only a series of MOFs are covered. So I think it would be better if they made use of the term MOF in the title instead.

I think the work would be of interest to the community although I think it would be less so to the wider field, given it is looking at a specific phenomenon in MOFs (or soft porous crystals).

The term soft porous crystals was first coined in a seminal 2009 review by Horike, Shimomura, and Kitagawa that appeared in *Nature Chemistry* [16]. In this review, they defined soft porous crystals as “porous solids that possess both a highly ordered network and structural transformability. They are bistable or multistable crystalline materials with long-range structural ordering, a reversible transformability between states, and permanent porosity.” Making the connection with the early classification of metal-organic frameworks (MOFs) or porous coordination polymers (PCPs) into three generations [17], Kitagawa and co-workers define soft porous crystals as third-generation MOFs, *i.e.*, “flexible or dynamic porous frameworks that reversibly respond to external stimuli, not only chemical but also physical.” Since then, the terms “soft porous crystals” and “flexible MOFs” have been used interchangeably to refer to these third-generation MOFs [3, 18, 19, 20].

As a result, as SPCs form a subclass of MOFs, we are confident that the term “soft porous crystals” is better suited to describe the scope of the work than the broader class of “metal-organic frameworks”, as we do not cover rigid MOFs in this work and as we cover four different examples of these SPCs. We have taken care, however, to properly define SPCs early in the introduction (page 2) to avoid any misinterpretation:

“Such phenomenon is here demonstrated to exist in soft porous crystals (SPCs) or flexible MOFs [21, 22], which show large-amplitude transitions between different (meta)stable phases while retaining their crystallinity [3, 16, 23].”

References

- [1] J. Keupp and R. Schmid, "Molecular Dynamics Simulations of the "Breathing" Phase Transformation of MOF Nanocrystallites," *ChemRxiv*, 2019. https://chemrxiv.org/articles/Molecular_Dynamics_Simulations_of_the_Breathing_Phase_Transformation_of_MOF_Nanocrystallites/8281082/1.
- [2] S. M. J. Rogge, L. Vanduyfhuys, A. Ghysels, M. Waroquier, T. Verstraelen, G. Maurin and V. Van Speybroeck, "A Comparison of Barostats for the Mechanical Characterization of Metal-Organic Frameworks," *J. Chem. Theory Comput.*, vol. 11, pp. 5583-5597, 2015.
- [3] L. Vanduyfhuys, S. M. J. Rogge, J. Wieme, S. Vandenbrande, G. Maurin, M. Waroquier and V. Van Speybroeck, "Thermodynamic Insight into Stimuli-Responsive Behaviour of Soft Porous Crystals," *Nat. Commun.*, vol. 9, p. 204, 2018.
- [4] P. P. Ewald, "Die Berechnung optischer und elektrostatischer Gitterpotentiale," *Ann. Phys.*, vol. 369, pp. 253-287, 1921.
- [5] C. Triguero, F.-X. Coudert, A. Boutin, A. H. Fuchs and A. V. Neimark, "Mechanism of Breathing Transitions in Metal-Organic Frameworks," *J. Phys. Chem. Lett.*, vol. 2, pp. 2033-2037, 2011.
- [6] Y. Sakata, S. Furukawa, M. Kondo, K. Hirai, N. Horike, Y. Takashima, H. Uehara, N. Louvain, M. Meilikhov, T. Tsuruoka, S. Isoda, W. Kosaka, O. Sakata and S. Kitagawa, "Shape-Memory Nanopores Induced in Coordination Frameworks by Crystal Downsizing," *Science*, vol. 339, pp. 193-196, 2013.
- [7] L. D. Landau and E. M. Lifschitz, *Statistical Physics, Part 1: Volume 5*, Butterworth-Heinemann, 1980.
- [8] S. M. J. Rogge, M. Waroquier and V. Van Speybroeck, "Reliably Modeling the Mechanical Stability of Rigid and Flexible Metal-Organic Frameworks," *Acc. Chem. Res.*, vol. 51, pp. 138-148, 2018.
- [9] J. E. Basconi and M. R. Shirts, "Effects of Temperature Control Algorithms on Transport Properties and Kinetics in Molecular Dynamics Simulations," *J. Chem. Theory Comput.*, vol. 9, pp. 2887-2899, 2013.
- [10] F.-X. Coudert, A. Boutin, M. Jeffroy, C. Mellot-Draznieks and A. H. Fuchs, "Thermodynamic Methods and Models to Study Flexible Metal-Organic Frameworks," *ChemPhysChem*, vol. 12, pp. 247-258, 2011.
- [11] S. M. J. Rogge, R. Goeminne, R. Demuyne, J. J. Gutiérrez-Sevillano, S. Vandenbrande, L. Vanduyfhuys, M. Waroquier, T. Verstraelen and V. Van Speybroeck, "Modeling Gas Adsorption in Flexible Metal-Organic Frameworks via Hybrid Monte Carlo/Molecular Dynamics Schemes," *Adv. Theory Simul.*, vol. 2, p. 1800177, 2019.
- [12] T. Kundu, M. Wahiduzzaman, B. B. Shah, G. Maurin and D. Zhao, "Solvent-Induced Control over Breathing Behavior in Flexible Metal-Organic Frameworks for Natural-Gas Delivery," *Angew. Chem., Int. Ed.*, vol. 58, pp. 8073-8077, 2019.
- [13] H. Miura, V. Bon, I. Senkovska, S. Watanabe, M. Ohba and S. Kaskel, "Tuning the Gate-Opening Pressure and Particle Size Distribution of the Switchable Metal-Organic Framework DUT-8(Ni) by Controlled Nucleation in a Micromixer," *Dalton Trans.*, vol. 46, pp. 14002-14011, 2017.
- [14] S. Krause, V. Bon, I. Senkovska, D. M. Töbrens, D. Wallacher, R. S. Pillair, G. Maurin and S. Kaskel, "The Effect of Crystallite Size on Pressure Amplification in Switchable Porous Solids," *Nat. Commun.*, vol. 9, p. 1573, 2018.
- [15] S. Wannapaiboon, A. Schneemann, I. Hante, M. Tu, K. Epp, A. L. Semrau, C. Sternemann, M. Paulus, S. J. Baxter, G. Kieslich and R. A. Fischer, "Control of Structural Flexibility of Layered-Pillared Metal-Organic Frameworks Anchored at Surfaces," *Nat. Commun.*, vol. 10, p. 346, 2019.

- [16] S. Horike, S. Shimomura and S. Kitagawa, "Soft Porous Crystals," *Nat. Chem.*, vol. 1, pp. 695-704, 2009.
- [17] S. Kitagawa and M. Kondo, "Functional Micropore Chemistry of Crystalline Metal Complex-Assembled Compounds," *Bull. Chem. Soc. Jpn*, vol. 71, pp. 1739-1753, 1998.
- [18] A. U. Ortiz, A. Boutin, A. H. Fuchs and F.-X. Coudert, "Anisotropic Elastic Properties of Flexible Metal-Organic Frameworks: How Soft are Soft Porous Crystals?," *Phys. Rev. Lett.*, vol. 109, p. 195502, 2012.
- [19] Z. Wang, N. Sikdar, S.-Q. Wang, X. Li, M. Yu, X.-H. Bu, Z. Chang, X. Zou, Y. Chen, P. Cheng, K. Yu, M. J. Zaworotko and Z. Zhang, "Soft Porous Crystal Based upon Organic Cages That Exhibit Guest-Induced Breathing and Selective Gas Separation," *J. Am. Chem. Soc.*, vol. 141, pp. 9408-9414, 2019.
- [20] F.-X. Coudert, "Soft Porous Crystals: Extraordinary Responses to Stimulation," *Bull. Jpn. Soc. Coord. Chem.*, vol. 73, pp. 15-23, 2019.
- [21] F.-X. Coudert, "Responsive Metal-Organic Frameworks and Framework Materials: Under Pressure, Taking the Heat, in the Spotlight, with Friends," *Chem. Mater.*, vol. 27, pp. 1905-1916, 2015.
- [22] F.-X. Coudert and A. H. Fuchs, "Computational Characterization and Prediction of Metal-Organic Framework Properties," *Coord. Chem. Rev.*, vol. 307, pp. 211-236, 2016.
- [23] A. Schneemann, V. Bon, I. Schwedler, I. Senkowska, S. Kaskel and R. A. Fischer, "Flexible Metal-Organic Frameworks," *Chem. Soc. Rev.*, vol. 43, pp. 6062-6096, 2014.

REVIEWERS' COMMENTS:

Reviewer #1 (Remarks to the Author):

Rogge et al. have carefully considered each of the reviewers' comments and revised the manuscript, accordingly. These revisions thoroughly accounted for in their response document answer the previously raised questions. I believe this manuscript should be published as is, subject to one minor change: the preprint of Keupp et al. has since been published as [10.1002/adts.201900117](https://doi.org/10.1002/adts.201900117).

Reviewer #1

Rogge et al. have carefully considered each of the reviewers' comments and revised the manuscript, accordingly. These revisions thoroughly accounted for in their response document answer the previously raised questions. I believe this manuscript should be published as is, subject to one minor change: the preprint of Keupp et al. has since been published as 10.1002/adts.201900117.

We would like to thank the reviewer for his/her careful reading of the manuscript and the positive evaluation. As the work of Keupp *et al.*, reference 8 in the revised manuscript, is now accepted and published online, we have adapted the reference as proposed by the reviewer.